# DDB1 engagement defines the selectivity of S656 analogs for cyclin K degradation over CDK inhibition

Céline Moison [1,6], Rodrigo Mendoza-Sanchez [1,6], Deanne Gracias[1,6], Doris A Schuetz [1], Jean-François Spinella[1], Simon Girard [1], Bounkham Thavonekham [1], Jalila Chagraoui[1], Aurélie Durand[1], Simon Fortier[1], Tara MacRae[1], Eric Bonneil[1], Yannick Rose[1], Nadine Mayotte[1], Isabel Boivin[1], Pierre Thibault [1,2], Josée Hébert [1,3,4,5], Réjean Ruel[1], Anne Marinier [1,2✉] & Guy Sauvageau [1,3,4,5✉]

## Abstract

In efforts to identify additional therapeutic targets for Acute Myeloid Leukemia (AML), we performed a high-throughput screen that includes 56 primary specimens tested with 10,000 structurally diverse small molecules. One specific hit, called S656 acts as a molecular glue degrader (MGD), that mediates the CRL4-dependent proteolysis of cyclin K. Structurally, S656 features a moiety that binds to the ATP binding site of cyclin-dependent kinases (CDKs), allowing the recruitment of the CDK12-cyclin K complex, along with a binding site for DDB1 bridging the CRL4 complex. Structure activity relationship studies reveal that minimal modifications to the dimethylaniline moiety of S656 improve its cyclin K MGD function over CDK inhibition by promoting DDB1 engagement. This includes full occupation of the DDB1 pocket, preferably with hydrophobic terminal groups, and cation-π interaction with Arg928. Additionally, we demonstrate that despite structural diversity, cyclin K degraders exhibit similar functional activity in AML which is distinct from direct CDK12 inhibition.

**Keywords** Molecular Glue Degrader; Cyclin K; CDK12; DDB1; Acute Myeloid Leukemia
**Subject Categories** Cancer; Pharmacology & Drug Discovery; Post-translational Modifications & Proteolysis

## Introduction

Acute Myeloid Leukemia (AML) is an aggressive disease with a high relapse rate and poor 5-year survival, despite initial responses to induction therapies. AML's heterogeneity means outcomes heavily depend on cytogenetics and mutational profiles, paving the way for the emergence of targeted therapies that exploit specific AML vulnerabilities. Such targeted therapies have gained significance over the last two decades (Cucchi et al, 2021; Totiger et al, 2023); however, developing safer and more effective therapies remains an unmet medical need. As part of the Leucegene initiative, our team has identified several novel therapeutic targets for this disease and developed preclinical compounds. Among these, we identified Mubritinib as an OXPHOS targeting compound most active in the NPM1c/FLT3-ITD subgroup (Baccelli et al, 2019), a copper ionophore molecule for *SF3B1*-mutated AMLs (Moison et al, 2024) and reported herein, a cyclin K molecular glue degraders (MGD) with broad anti-AML activity.

Cyclin K is a regulatory protein for several cyclin-dependent kinases including CDK12, CDK13 and CDK9. Through phosphorylation of Ser2 of the C-terminal domain of RNA polymerase II, the CDK12-cyclin K protein complex regulates transcription elongation, focusing on lengthy, exon-rich genes, such as DNA damage response genes (Bartkowiak et al, 2010; Edwards et al, 1998; Kohoutek and Blazek, 2012; Blazek et al, 2011). Both CDK12 and cyclin K are essential, showing co-dependencies in DepMap genome-wide screens (Tsherniak et al, 2017), highlighting their functional connection. CDK12 has been explored as a cancer treatment target (Wu et al, 2023; Tang et al, 2022; Chou et al, 2020; Savoy et al, 2023), typically through direct inhibition of its ATP-binding site. However, a recent paradigm shift occurred when MGDs targeting cyclin K for degradation were identified (Słabicki et al, 2020; Lv et al, 2020; Mayor-Ruiz et al, 2020; Dieter et al, 2021), offering an alternative approach to disrupt the CDK12-cyclin K complex. An unanswered question, partially addressed in a recent study (Kozicka et al, 2024), is whether these two approaches are biologically equivalent.

Targeted protein degradation by MGDs has become a powerful strategy for expanding the range of targetable proteome. This approach offers the potential to address proteins lacking enzymatic activity or well-defined domains, which as for cyclin K, were previously considered 'undruggable'. Molecular glues are monovalent small molecules that enhance or create protein-protein interaction, leading to the ubiquitination and degradation of target

[1]Institute for Research in Immunology and Cancer, Université de Montréal, Montreal, Quebec, Canada. [2]Department of Chemistry, Université de Montréal, Montreal, Quebec, Canada. [3]Institut universitaire d'hémato-oncologie et de thérapie cellulaire, Maisonneuve-Rosemont Hospital, Montreal, Quebec, Canada. [4]Quebec Leukemia Cell Bank, Maisonneuve-Rosemont Hospital Research Center, Montreal, Quebec, Canada. [5]Department of Medicine, Faculty of Medicine, Université de Montréal, Montreal, Quebec, Canada. [6]These authors contributed equally: Céline Moison, Rodrigo Mendoza-Sanchez, Deanne Gracias. ✉E-mail: anne.marinier@umontreal.ca; guy.sauvageau@umontreal.ca

proteins when an E3 ubiquitin ligase is involved. These MGDs, therefore, rely on a tripartite interaction between two proteins with complementary interfaces and a small molecule. In this arrangement, the protein of interest is precisely positioned, typically through its binding to a substrate adaptor of E3 ligase complexes, to undergo poly-ubiquitination.

CR8 was the first reported MGD that targeted cyclin K (Słabicki et al, 2020). In this elegant work, Ebert and collaborators demonstrated the contribution of DDB1 as the Cullin-Ring Ligase Complex 4 (CRL4) substrate receptor to which CDK12 binds to, and that chemical alteration of surface-exposed moieties can confer gain-of-function glue properties to a CDK inhibitor. Subsequent studies identified other cyclin K MGDs (Lv et al, 2020; Mayor-Ruiz et al, 2020; Dieter et al, 2021; Sano et al, 2023; Zhang et al, 2024), including a recent comprehensive crystallographic study revealing key features of such degraders (Kozicka et al, 2024). Despite their structural diversity, MGDs operate similarly, interacting with both DDB1 and the kinase domain of CDK12, leading to the efficient poly-ubiquitination and degradation of cyclin K by the proteasome.

Herein, we report the identification, optimization, and mechanistic exploration of S656, a cyclin K MGD with selective anti-AML activities while sparing normal CD34[+] cells. Through focused structure activity relationship and genetic studies, we developed specific cyclin K degrader molecules and confirmed the structural elements that best distinguish cyclin K degraders from CDK inhibitors. Moreover, we demonstrated that cyclin K MGDs and CDK12/13 inhibitors produce specific and distinct cellular fingerprints establishing that both series of compounds are not equivalent.

# Results

## High-throughput screening identifies S656 as a selective anti-proliferative molecule in AML

As recently described, our group conducted a high-throughput screening (HTS) assay assessing the response of 56 primary AML specimens to a library of 10,000 compounds (Moison et al, 2024). This diverse small molecule library encompassed compounds with a wide range of structural diversity (Bristol-Myers Squibb), including clinical-grade compounds. Primary AML specimens representing the biological diversity of the disease, along with normal hematopoietic cells (CD34[+] cord blood cells), were exposed to a 6-day single-dose compound treatment (1 µM) and viability was assessed using the CellTiterGlow luminescent assay (Appendix Fig. S1A–C). A profile of inhibition—or fingerprint—was generated for each small molecule across the 56 primary AML specimens tested. From this analysis, we compiled a shortlist of small molecules that exhibited a high percentage of growth inhibition only in a subset of primary specimens while excluding compounds with general cytotoxicity (Fig. 1A). Among these candidates, 12 were validated in cell lines and primary specimens and selected based on their chemotypes. Here we present the identification and characterization of the S656 hit (Fig. 1B) which showed more than 70% growth inhibition in 10 out of 56 primary AML specimens after exposure to 1 µM of the molecule (Fig. 1C; Dataset EV1). Importantly, S656 exhibited low activity against CD34[+] cord blood cells. Of note, highly sensitive AML specimens spanned various

AML subgroups, underscoring the broad activity of S656 across genetically diverse AMLs (Fig. 1D).

To better stratify S656's response in AML, we tested S656 potency in dose-response assays and determined half-maximal inhibitory ($IC_{50}$) concentration in a panel of 157 genetically diverse primary AMLs (Fig. 1E; Appendix Fig. S1D; Dataset EV1). A bimodal distribution of the $IC_{50}$ values was obtained (Fig. 1F), separating more sensitive from more resistant AMLs. While PML-RARa, RUNX1-RUNX1T1, NUP98-NSD1, inv(16), and Monosomy 5/5q-/7/7q- were more prone to present sensitivity (Fig. 1G), no significant association was found between S656's response modes and cytogenetic groups, mutational status or clinical features. Although the sample size of some of these variables limits the power of our analysis, this suggests that S656's sensitivity is rather driven by mechanisms that are independent of the tested features.

## S656 mediates cyclin K degradation through the cullin RING ubiquitin ligase complex

To gain insight into the mode of action of S656, we performed a genome-wide CRISPR/Cas9 loss-of-function screen using the OCI-AML1 cells stably transduced with the EKO sgRNA library (Bertomeu et al, 2018). Cells were exposed to 1 µM of S656 compound, for 10 doublings over two weeks. Subsequent enrichment and depletion of guide RNAs were assessed by RNA sequencing to identify synthetic rescue and synthetic lethal interactions (Dataset EV2). Among the top hits that provided synthetic rescue in the presence of S656, we found several components of the CRL4 complex, a multisubunit protein complex responsible for ubiquitinating and subsequently degrading target proteins (Fig. 2A,B). This includes *CUL4A/B* (cullin), *RBX1* (RING), *DDB1* (substrate adaptor) and *UBE2G1* (E2 ligase) genes. We also observed that disruptions in CRL regulatory genes such as *GLMN* and *SENP8* were associated with resistance to S656. Notably, no known substrate-specific receptor of the DDB1-CRL4 complex (DCAF) conferred resistance to S656 when Cas9-edited. Supporting the genetic evidence of CRL4-dependent mediated toxicity, S656 potency was decreased in the presence of MLN4924, a neddylation inhibitor that impairs CRL complex activities (Fig. 2C). Such mechanism of action is reminiscent of MGDs, which create or stabilize the interaction between a neo-substrate and a protein of one of the CRL complexes, leading to CRL-mediated proteolysis.

A total proteomic analysis of OCI-AML5 cells identified cyclin K as the most depleted protein upon short-time exposure to S656 (Fig. 2D; Dataset EV3), suggesting that it is targeted by the molecule. This finding was further validated by western blot analysis, which revealed a strong depletion of cyclin K in 3 primary AML specimens following S656 treatment (Appendix Fig. S2A). Using a reporter system in which cyclin K is ectopically expressed in fusion with eGFP (Słabicki et al, 2020), we consistently observe a strong reduction in cyclin $K_{eGFP}$ levels after a 3-h incubation with S656 (Fig. 2E; Appendix Fig. S2B–D). Other previously reported cyclin K degraders such as HQ461, CR8, and SR4835 provided comparable results while CDK inhibitors (THZ531 and SNS-032) maintained cyclin $K_{eGFP}$ levels in this system. In the OCI-AML5 cell line, we demonstrated a dose-dependent cyclin K depletion after treatment with either S656 or HQ461 (Fig. 2F). CDK12, the binding partner to cyclin K was also destabilized by this treatment.

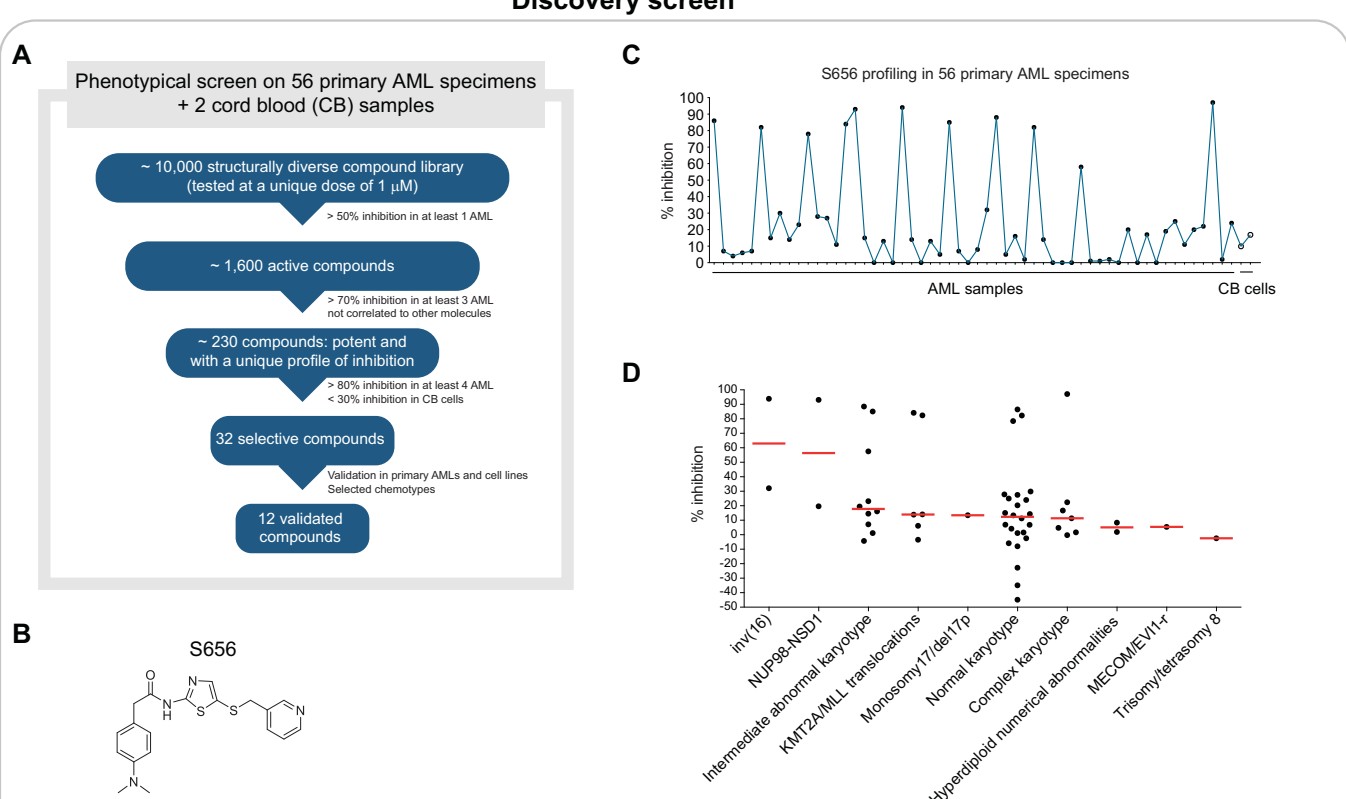

**Figure 1. High-throughput screening identifies S656 as a selective anti-proliferative molecule in AML.**

(A) Selection of candidate hits from the discovery screen on primary AML samples. (B) Chemical structure of the S656 hit. (C) Inhibitory profile of S656 compound across 56 primary AML specimens and 2 cord blood (CB) samples. Percentage of inhibition at 1 μM, after 6 days incubation, normalized to DMSO control treatment. (D) Dot plot distribution of S656 associated percentage of inhibition (at 1 μM) across the different AML specimen subgroups. Normalized to DMSO control treatment, median is represented in red ($n = 56$ primary specimens). (E) AML subtype classification of the 157 primary specimens used in validation screen. (F) Bi-modal distribution of S656 $IC_{50}$ values obtained in the validation screen (dose-response, 6 days incubation, normalized to DMSO control). (G) Dot plot distribution of S656 $IC_{50}$ values across primary AML specimens. Blue and orange dots represent more sensitive and more resistant specimens respectively as defined in the bi-modal distribution in (F) ($n = 157$ primary specimens). See also Appendix Fig. S1.

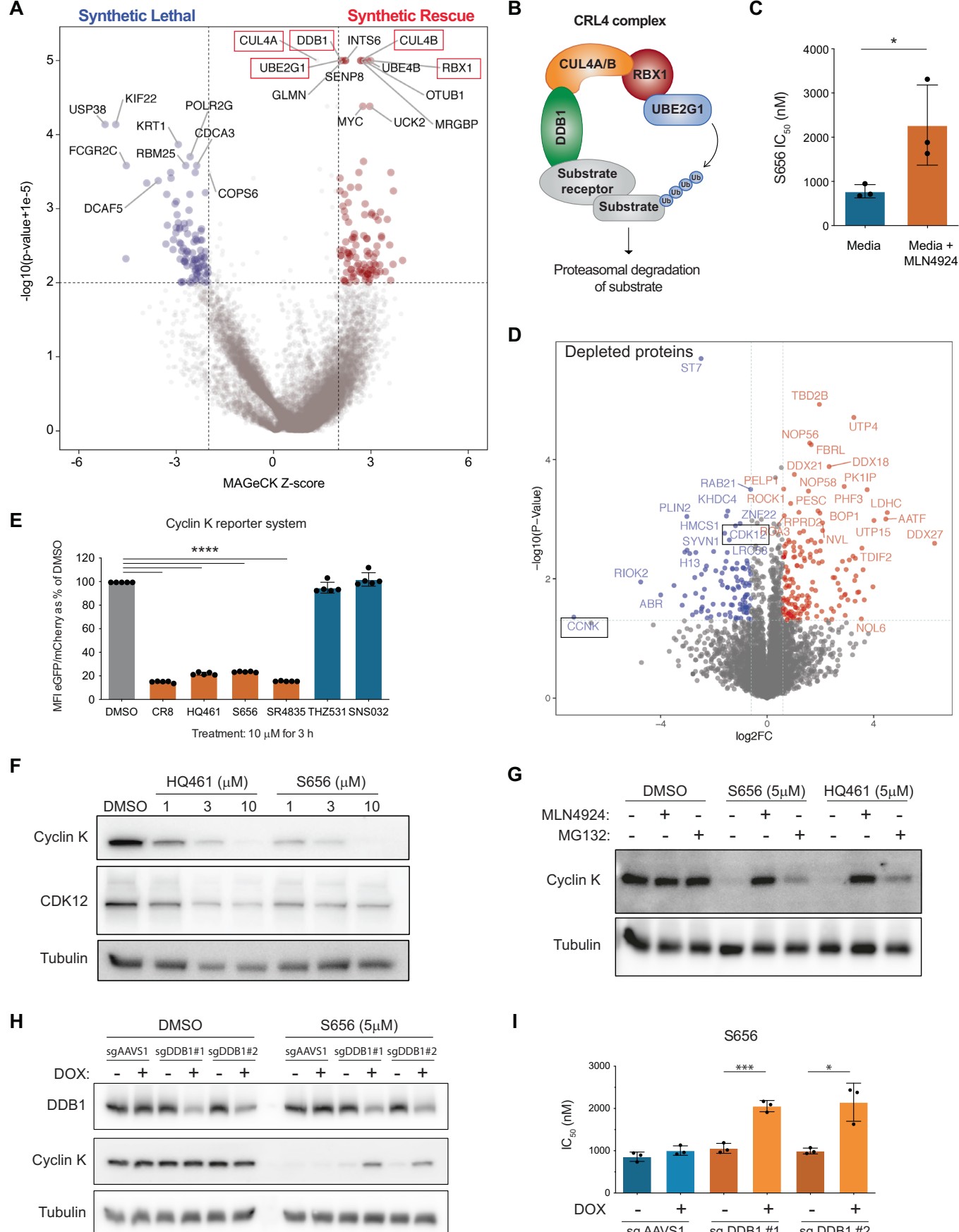

◀ **Figure 2. S656 mediates cyclin K degradation through the cullin RING ubiquitin ligase complex.**

(A) Volcano plot representing results of whole genome CRISPR/Cas9 screen performed in EKO OCI-AML1 cells during exposure to 1 μM of S656 compound ($n = 1$, around 10 sgRNAs per gene). (B) Schematic representation of the Cullin4-RING E3 ubiquitin ligase (CRL4) complex. (C) Dose response experiment to determine $IC_{50}$ values of S656 in OCI-AML5 cells, in regular media or supplemented by 25 nM of the neddylation inhibitor MLN4924 to prevent CRL-mediated proteolysis. $IC_{50}$ values were determined after 4 days of incubation (mean $+/-$ SD, $n = 3$, biological replicates, $t$ test, $P$ value $= 0.0478$). (D) Quantitative proteome-wide mass spectrometry analysis performed in OCI-AML5 cells exposed to 8 μM of S656 for 5 h (CCNK = cyclin K, $n = 3$, biological replicates). (E) Cyclin K degradation assessment by measuring the mean fluorescence intensity (MFI) of cyclin $K_{eGFP}$ over mCherry by flow cytometry. OCI-AML5 G7 clone was treated for 3 h with 10 μM of the indicated compounds. Results are normalized to fluorescence in DMSO-treated cells (mean $+/-$ SD, $n = 5$, biological replicates, $t$ test, $P$ value $< 0.0001$). (F) Immunoblot analysis of cyclin K and CDK12 protein levels in OCI-AML5 cells, after exposure to increasing concentrations of HQ461 or S656 for 6 h. Tubulin is used as a loading control. (G) Immunoblot analysis of cyclin K protein levels after exposure to HQ461 or S656 (5 μM for 5 h). Where indicated, OCI-AML5 cells were pre-treated 1 h with 500 nM of MLN4924 or MG132. (H) Immunoblot analysis of DDB1 and cyclin K protein levels after exposure to S656 (5 μM for 5 h) in OCI-AML5 Cas9 cells expressing inducible ($+$DOX) sgRNAs targeting *DDB1* or control region *AAVS1*. (I) Dose response experiment to determine S656 $IC_{50}$ values in inducible ($+$DOX) OCI-AML5 Cas9 cells expressing two different sgRNAs targeting *DDB1* gene or *AAVS1* control region (mean $+/-$ SD, 4 days incubation, $n = 3$, biological replicates, $t$ test, $P$ value $= 0.0006$ and $0.0118$ in sgDDB1 #1 and #2, respectively). See also Appendix Fig. S2. Source data are available online for this figure.

MLN4924, and to a lesser extent, MG132 (proteasome inhibitor) treatment prevented cyclin K degradation by S656 (Fig. 2G), demonstrating that cyclin K is actively degraded through CRL-mediated proteolysis. Additionally, Cas-9 mediated knocking down of *DDB1* partly rescued cyclin K degradation by S656 (Fig. 2H) and expectedly reduced the potency of this compound (Fig. 2I), demonstrating that S656 cytotoxicity is dependent on its capacity to degrade cyclin K. Similar results were obtained with known cyclin K MGDs (HQ461, CR8 and SR4835) (Appendix Fig. S2E). Likewise, the downregulation of *CUL4A* or *CUL4B* by shRNAs impaired S656 and known cyclin K MGDs potency, confirming the CRL4 complex requirement for the associated cytotoxicity to this class of molecules (Appendix Fig. S2F,G). In contrast, potency of SNS-032 or THZ531, a selective and covalent CDK12/13 inhibitor, remained unaffected by *DDB1* or *CUL4A/B* depletion (Appendix Fig. S2E,F).

## S656 suppresses expression of DDR genes and induces DNA lesions

As part of the cyclin K-CDK12 complex which regulates transcriptional elongation, cyclin K depletion induced by S656 or HQ461 expectedly leads to reduced levels of RNA polymerase II phospho-ser2 (Fig. 3A). Notably, this reduction can be fully or partially rescued in the presence of MLN4924 or MG132, respectively. Given that the cyclin K-CDK12 complex has previously been reported to affect the expression of DNA damage response (DDR) genes (Blazek et al, 2011), we next compared the relative expression levels of selected DDR transcripts namely, *BRCA1-2*, *ATR*, *BLM*, *ERCC4*, *BARD1* and *RAD51* in response to S656 and THZ531 (CDK12/13 inhibitor). We observed a significant decrease in the levels of all these transcripts after 4 h exposure to both compounds (Fig. 3B; Appendix Fig. S3A). Aligning with the known role of CDK12 in regulating G1/S progression (Chirackal Manavalan et al, 2019), we observed that cells accumulated in the G1-cell cycle phase after 24 h exposure to S656 (Fig. 3C). Concurrently, this cell-cycle arrest was associated with a dose-dependent increase in apoptosis, as indicated by Annexin V/PI staining (Fig. 3D; Appendix Fig. S3B), and the accumulation of gH2AX foci, indicative of unrepaired DNA lesions (Fig. 3E). According to previous observations with CR8 molecule (Delehouzé et al, 2014; Bettayeb et al, 2010), short-term exposure to S656 also induces a substantial depletion of the anti-apoptotic Mcl-1 protein and c-MYC oncogene (Fig. 3F), which may contribute to the cellular effects observed with exposure to S656.

## Cyclin K degraders and CDK12/13 inhibitors exhibit distinct fingerprints in primary AMLs

We showed that S656 induces DNA lesions and cell cycle aberrations similar to those caused by the covalent CDK12/13 inhibitor THZ531. We thus explored whether the biological response to cyclin K degradation is equivalent to the direct inhibition of CDK12/13 by exploiting our previously reported compound-directed fingerprints in primary AML (Baccelli et al, 2017). By conducting a comparative analysis of the potency and inhibitory profiles of S656, cyclin K MGDs and CDK inhibitors across a panel of 40 primary AML specimens (Fig. 4A), we clearly showed that all cyclin K degraders exhibited a common fingerprint and clustered together. Despite their structural diversity and differences in potency (Fig. 4B), they shared a high correlation in $IC_{50}$ variation across the AML panel (Fig. 4C). Strikingly, the fingerprint generated by THZ531 was distinct and showed no correlation with any cyclin K degrader profiles. Instead, it correlated with SNS-032 (correlation of 0.78), a potent inhibitor of CDK2/7/9 (Misra et al, 2004; Heath et al, 2008). Together, these results suggest that the cellular response to cyclin K degradation is not equivalent to direct inhibition of CDK12/13 and that different mechanisms of action are at play.

## S656 mediates interaction between CDK12 and DDB1

Crystallographic studies have shown how cyclin K degraders stabilize the formation of the protein-complex of CDK12 and DDB1 due to distinct structural elements interacting independently with the kinase domain of CDK12 as well as with the surface of DDB1 (Słabicki et al, 2020; Lv et al, 2020; Mayor-Ruiz et al, 2020; Dieter et al, 2021). The aminothiazole core present in S656 has been described to bind to the hinge region of CDKs (Misra et al, 2004; Kim et al, 2002). Accordingly, the structural pattern of a H-bond donor adjacent to a H-bond acceptor, right next to an aromatic CH, a triad present in the aminothiazole core, represents the hinge-binder motif.

Using Glide (Schrödinger Release 2022, LLC), we perform induced fit docking of S656 in X-ray PDB: 6TD3 (Słabicki et al, 2020), which contains the known MGD CR8 bound to the CDK12-DDB1 complex (Kozicka et al, 2024). CR8 forms a hydrogen bond network with Met816 and Glu814 of CDK12's hinge region, while forming a cation-π interaction with Arg928 of DDB1, as crucial interactions (Słabicki et al, 2020; Kozicka et al, 2024) (Appendix

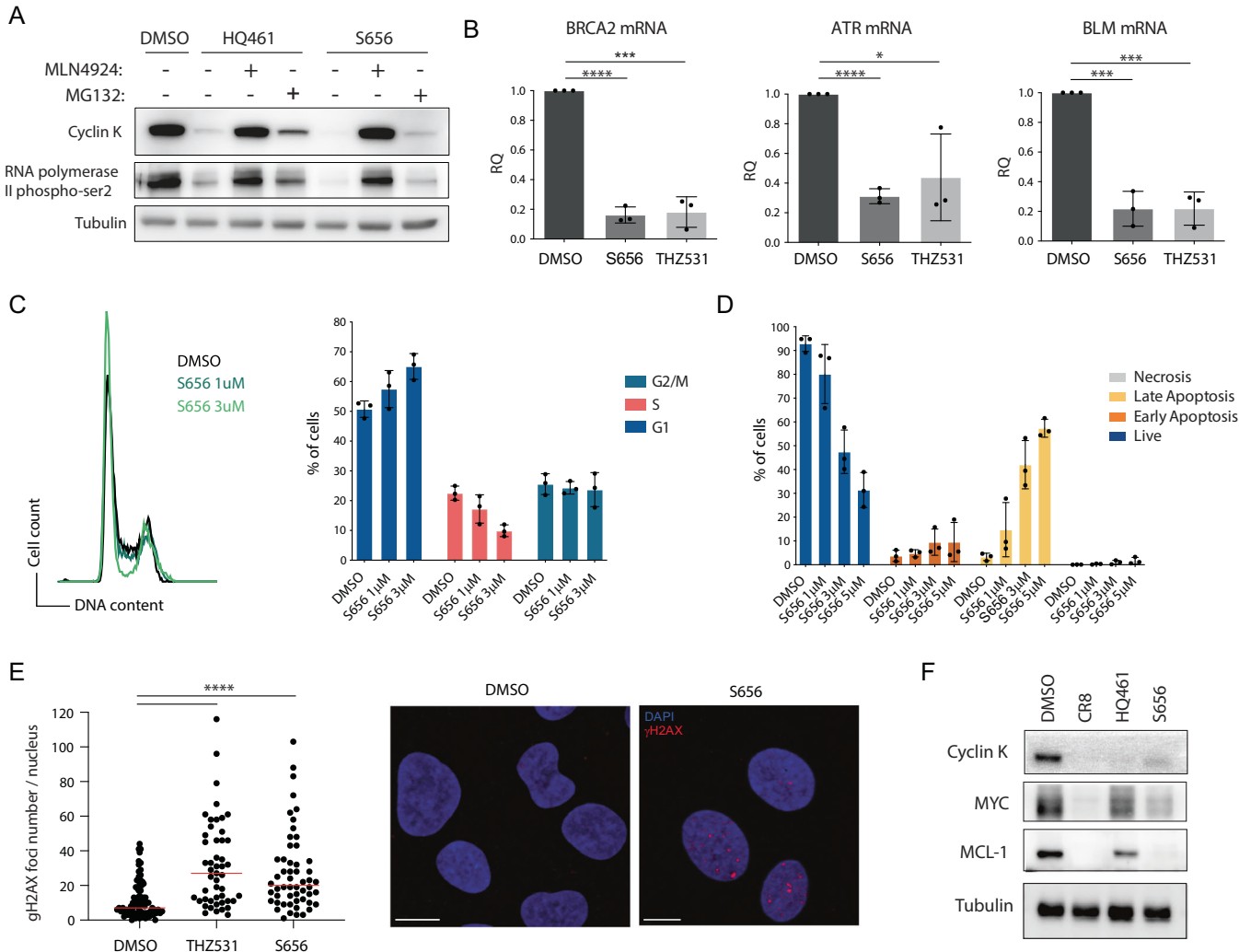

**Figure 3. S656 suppresses expression of DDR genes and induces DNA lesions.**

(A) Immunoblot analysis of cyclin K and RNA polymerase II phospho-ser2 protein levels in OCI-AML5 cells, after exposure to HQ461 or S656 (5 µM for 5 h). Where indicated, cells were pre-treated 1 h with 500 nM of MLN4924 or MG132. Tubulin is used as a loading control. (B) Monitoring of *BRCA2*, *ATR* and *BLM* mRNA expression by qPCR in OCI-AML5 cells treated 4 h with 5 µM of S656 or 200 nM of the CDK12/13 inhibitor THZ531. Normalized to *HPRT* (mean +/− SD, n = 3, biological replicates, *t* test, P value < 0.0001 (DMSO vs S656) and P value = 0.0002 (DMSO vs THZ531) in *BRCA* mRNA, P value < 0.0001 (DMSO vs S656) and P value = 0.0292 (DMSO vs THZ531) in *ATR* mRNA, P value = 0.0003 (DMSO vs S656) and P value = 0.0003 (DMSO vs THZ531) in *BLM* mRNA). (C) Cell cycle profile (left) and quantification of the percentage of OCI-AML5 cells in cell cycle phases (right, mean +/− SD, n = 3, biological replicates), 24 h after exposure to S656. (D) Quantification of cell death by Annexin V/PI staining of OCI-AML5 cells treated for 24 h with increasing concentrations of S656 (mean +/− SD, n = 3, biological replicates). (E) U2OS cells were treated for 24 h with DMSO, 1 µM of THZ531 or 2 µM of S656, fixed with PFA and immunostained with γH2AX antibody. Quantification of the number of γH2AX foci per nucleus is presented (left, median is depicted in red, n = 3, biological replicates, *t* test, P value < 0.0001) with representative images (right, red: γH2AX, blue: DAPI, scale bar: 10 µm). (F) Immunoblot analysis of cyclin K, MYC and MCL-1 protein levels in OCI-AML5 cells after exposure to indicated cyclin K MGDs (5 µM for 4 h). Tubulin is used as a loading control. See also Appendix Fig. S3. Source data are available online for this figure.

Fig. S4A,B). Notably, this interaction pattern is consistent across all other publicly available crystal structures. Retrieved docking poses of S656 in 6TD3 (Fig. 5A,B) suggest that S656 retains the H-bond network with Met816 and the aromatic H-bond to Glu814 backbone carbonyl, with an additional hydrogen bond involving CDK12's Lys756. The most prominent interaction of S656 with DDB1 is the cation-π interaction established between Arg928 and the benzene moiety of the molecule. Critically, we observed that the pocket on DDB1 is filled and van der Waals interactions are established. Another important structural feature of degraders is

the methylene linker, present in S656, which accounts for the L-shaped conformation of the molecule. It seems structurally essential for cyclin K MGDs, as its removal, as showcased in S656 analog UOM-005845, abolishes this function (Fig. 5C).

To experimentally support a direct interaction between S656 and CDK12, we observed that S656 inhibited the kinase activity of CDK12-cyclin K in vitro (Fig. 5D). We further showed that pre-incubating cells with the CDK12/13 covalent inhibitor, THZ531, rescued S656-dependent cyclin K degradation (Fig. 5E). A similar result was observed with both HQ461 and CR8, implying that the

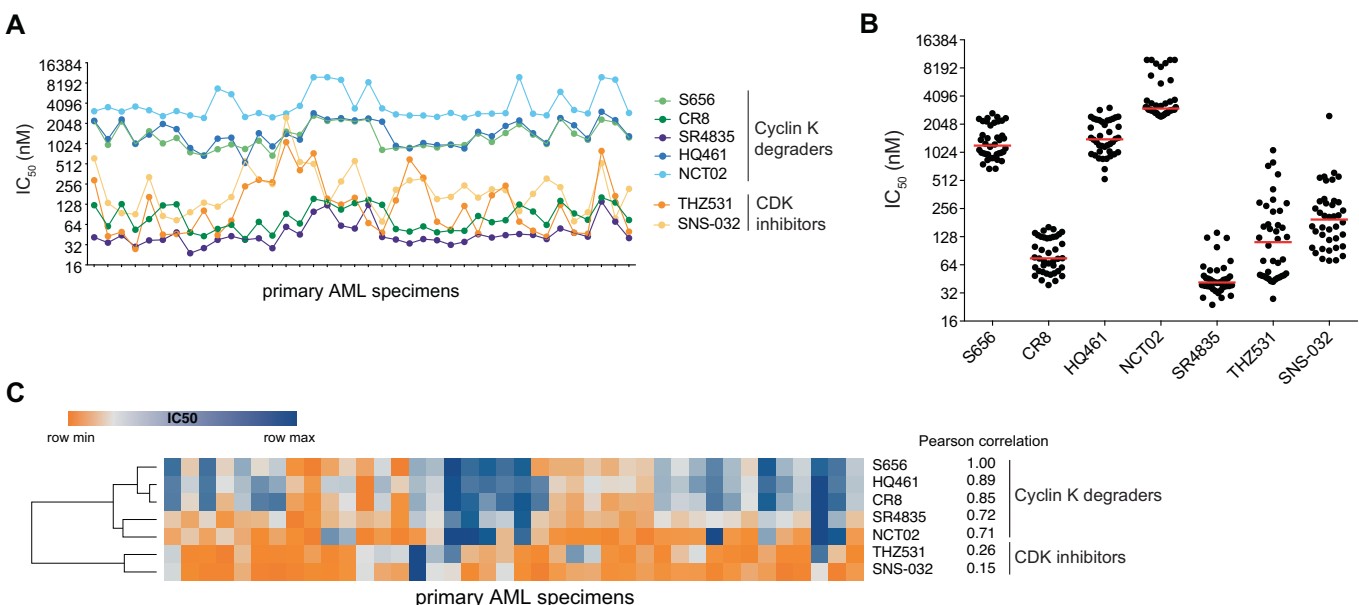

**Figure 4. Cyclin K degraders and CDK12/13 inhibitor show distinct fingerprints on primary AMLs.**

(A) Inhibitory profile of S656, CR8, SR4835, HQ461, NCT02, THZ531 and SNS-032 compounds across 40 primary AML specimens in dose response assays. IC$_{50}$ values were determined after 6 days of incubation and normalized to DMSO control. (B) Dot plot summarizing the IC$_{50}$ values and medians (red) obtained in the panel of 40 primary AMLs. (C) Heatmap representation and hierarchical clustering of IC$_{50}$ values from (A). Correlation between S656 and the other molecules is indicated. Relative color scheme uses the minimum and maximum IC$_{50}$ values in each row. Source data are available online for this figure.

cyclin K degraders bind to CDK12, and this engagement was essential for S656-mediated cyclin K degradation. Additionally, we demonstrated that S656 efficiently induced an interaction between CDK12 and DDB1 in live cells as monitored by the NanoLuc Binary Technology assay (Fig. 5F). The CDK12-DDB1 interaction occurred within minutes, in a dose-dependent manner upon incubation with S656 and other cyclin K MGDs but was not detectable in the presence of THZ531 (Appendix Fig. S4C).

Subsequently, we performed pull-down assays followed by quantitative mass spectrometry (MS) using a functionalized probe (UOM-005790) derived from S656 and its corresponding negative control (UOM-005839) derived from the inactive analog UOM-005628 (Fig. 5G). Strikingly, only a single atom (N versus C–H) in the aminothiazole that interacts in the CDK12 ATP binding pocket distinguishes S656 and UOM-005628 resulting in a non-degrading molecule. We observed that the H-bond network between UOM-005628 and CDK12's hinge region is disrupted due to the rotation of the thiadiazol, exposing the sulfur atom facing the hinge region (Fig. 5H; Appendix Fig. S4D), thus preventing the formation of crucial hinge-interactions. Binding pose Metadynamics (BPMD)—an enhanced sampling method (10 × 10 ns)—confirmed that compound S656 is significantly more stable in the CDK12-DDB1 binding site than UOM-005628 (Appendix Fig. S4E). Importantly, the UOM-005790 probe specifically identified cyclin K and several CDKs, including CDK12, as direct interactors (Fig. 5I; Dataset EV4). Interactors were competed out in the presence of excess S656 and were not identified using the negative control probe UOM-005839. As anticipated, DDB1 was not enriched using the UOM-005790 probe (Appendix Fig. S4F) as the PEG linker was incorporated in place of the DDB1 binding element.

## Hit compound S656 displays both cyclin K degrader function and CDK inhibitory activity

CDK inhibitors often exhibit poor selectivity due to the high conservation of residues in the active site of these enzymes. Accordingly, CR8 which binds to the kinase domain of CDK12, inhibits a wide range of CDKs (Delehouzé et al, 2014; Bettayeb et al, 2010). Similarly, S656, as a non-optimized hit, displays in vitro inhibitory activity for CDK1 and 9 (Dataset EV5). We thus employed a shRNA-based approach to dissect the contribution of CDK inhibition in S656, CR8, HQ461, SR4835 and SNS-032 anti-AML proliferation effects. In these experiments, we assessed the potency of selected molecules in OCI-AML5 cells expressing shRNA targeting *CDK1, 2, 4, 7,* and *9* as well as *cyclin K*. The IC$_{50}$ values in response to cyclin K MGDs—CR8, SR4835 and HQ461—were specifically and significantly reduced with *cyclin K* downregulation, but not by downregulation of any of the tested CDKs (Fig. 6A; Appendix Fig. S5). This suggests that cyclin K degradation is the primary mechanism by which these molecules inhibit OCI-AML5 proliferation. As a proof-of-concept, the proliferation of OCI-AML5 cells exposed to SNS-032 was significantly affected only in *CDK9*-downregulated cells. S656 was unique among these molecules, as its potency was significantly enhanced not only in *cyclin K* depleted cells but also in cells expressing shRNAs against *CDK1* or *CDK9*. These data suggest that S656 possesses CDK inhibitory activities in addition to its cyclin K degrader function.

## Dissociation of S656's degrader function from CDK inhibitory activity

The lack of selectivity in S656 prompted us to synthesize a small library of compounds through a focused SAR study to obtain a selective cyclin

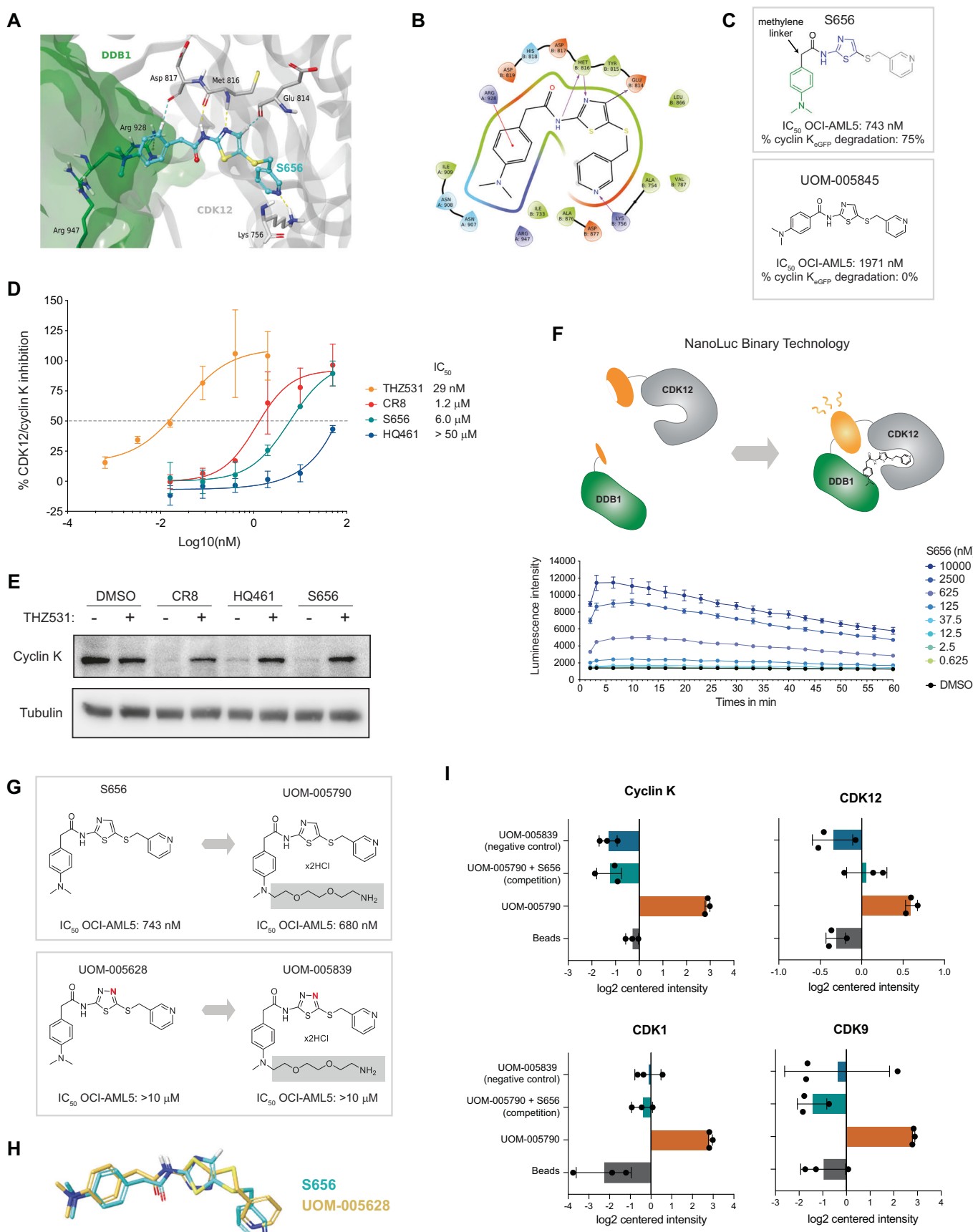

Figure 5. S656 mediates interaction between CDK12 and DDB1.

(A) 3D representation of S656 (turquoise, stick representation) bound to CDK12 (grey ribbons) and DDB1 (green surface representation). H-bonds are depicted as yellow dashed line and aromatic H-bonds are shown as turquoise dashed lines. Met816 provides two backbone interactions, a H-bond donor, and a H-bond acceptor, respectively. The CH on the thiazole core participates in an aromatic H-bond with Glu814 backbone carbonyl. S656's pyridine ring also engages in a H-bond with Lys756. The phenyl ring of S656 establishes a cation-$\pi$ interaction with Arg928 on DDB1, shown in green dashed lines and an aromatic H-bond to Asp817 on CDK12. (B) 2D interaction diagram of S656 bound to CDK12 and DDB1, in which purple arrows represent H-bonds and the red line stands for a cation-$\pi$ interaction. (C) Values of IC$_{50}$ in proliferative assay (OCI-AML5, 4 days incubation) and percentage of cyclin K$_{eGFP}$ degradation is displayed for S656 compound and its linear analog UOM-005845 (missing the methylene linker). Structural features of S656 are highlighted in different colors: methylene linked moiety (referred as "left-hand side") of the molecule in green, aminothiazole core in blue and the "right-hand side" in grey. (D) In vitro inhibition of CDK12/Cyclin K activity by increasing concentrations of indicated molecules (mean +/− SD, $n = 3$, biological replicates). THZ531 is used as a positive control. (E) Immunoblot analysis of cyclin K protein levels after exposure to CR8, HQ461 or S656 (5 μM for 5 h). Where indicated, OCI-AML5 cells were pre-treated 1 h with 5 μM of THZ531. (F) Schematic representation (top) of the NanoLuc Binary Technology used to monitor the interaction between CDK12 fused to the long bait and DDB1 fused to the short bait. Luminescence intensity (bottom) generated by the reconstitution of the NanoLuc was monitored over 1 h right after the addition of S656 compound at increasing concentrations in HEK293 cells (mean +/− SD, $n = 4$, biological replicates). (G) Structure of the pull-down probes UOM-005790 and UOM-005839 derived from S656 and UOM-005628 respectively (carries an additional nitrogen highlighted in red). Probes were designed by functionalizing the dimethylaniline moiety with a PEG linker (grey) which was then immobilized in beads. (H) Overlay of compounds UOM-005628 (yellow, stick representation) and S656 (turquoise, stick representation), when bound to CDK12 and DDB1 (proteins not shown). The rotation of the thiadiazole in UOM-005628 when compared to the thiazole in S656, exhibits a different binding motif towards the kinase hinge region. (I) Graphical representation of the pull-down enrichments obtained for the indicated proteins (analyzed using the R package DEP). Pull-down assays were performed using the UOM-005790 probe in absence or presence of excess S656 (competition condition), the negative control probe UOM-005839 or beads with no probes in OCI-AML5 total protein extracts (mean +/− SD, $n = 3$, biological replicates). See also Appendix Fig. S4. Source data are available online for this figure.

K degrader, while abolishing CDK-related cytotoxic activities. We assessed the analogs' potency in *CUL4A*-depleted OCI-AML5 cells and in the stability reporter system for cyclin K$_{eGFP}$ (Dataset EV5). Among the 54 analogs generated (Appendix Fig. S6), we observed that some of them lost the ability to degrade cyclin K while retaining anti-proliferative activities (Fig. 6B), suggesting a different target engagement. Interestingly, we observed that when efficiency of cyclin K degradation dropped below 65% in the reporter assay, anti-proliferative activity of the related molecules is no longer *CUL4A*-dependent (Fig. 6C), indicating that poor ability to degrade cyclin K translates into an alternative mechanism of action of these molecules. In our assay, we therefore established that a minimal degradation efficiency of 65% would be the requirement to qualify a molecule as an effective degrader. As observed for S656, most of the analogs possess in vitro CDK1 and/or 9 inhibitory activities (Dataset EV5) suggesting that those molecules, which lost cyclin K degrader ability, may function as CDK inhibitors.

We then hypothesized that only those molecules capable of stabilizing the CDK12-DDB1 complex through molecular interactions with both proteins behave as cyclin K MGDs, while molecules that cannot establish interaction with DDB1 will instead mediate cytotoxicity through CDK inhibition (see model in Fig. 6D).

## Minimal modifications in the DDB1-interacting surface optimize cyclin K degrader function

By assessing the potency of the 54 analogs in cells with reduced CDKs and cyclin K levels (as shown in Fig. 6A), and measuring their ability to degrade cyclin K$_{eGFP}$, we gained a thorough understanding of the biological activity and selectivity of our compound series. As a result, we categorized S656 analogs into different classes based on their cellular selectivity as represented in the heatmap of Fig. 7A. Class I clusters 8 compounds which are effective cyclin K MGDs (decreased IC$_{50}$ in *cyclin K* depleted cells only, and >65% degradation of cyclin K$_{eGFP}$), while Class II comprises molecules which have lost the ability to degrade cyclin K but retain CDK-related cytotoxicity. Analogous to S656, 9 compounds were grouped in a mixed Class I–II category,

displaying mild to high cyclin K degradation ability along with CDK-related activities.

To build the SAR, we first modified the dimethylaminophenyl acetate moiety of S656 (modifications highlighted in green in Fig. 7A and Appendix Fig. S6), which protrudes CDK12's ATP binding pocket and establishes interaction with the surface of DDB1 (*vide infra*). Class I comprises mostly compounds with two adjacent or fused aromatic rings, either a phenyl-heteroalkyl (UOM-005608), bi-aryl (UOM-005636) or naphthalene (UOM-005428). These observations are consistent with recent efforts regarding the deconvolution of the SAR for this class of compounds (Kozicka et al, 2024; Thomas et al, 2024). However, the aliphatic *p*-tolyl-pyrrolidine (UOM-005608) analog of S656 was the only heterocycle of its kind that grouped in Class I, and displayed a similar potency in OCI-AML5 cells when compared to UOM-005636. This highlights that aliphatic groups can also favor the recruitment of DDB1 in a selective manner. However, minimal modifications of Class I compounds affect their selectivity and shifts the compounds to a mixed Class I–II. Indeed, the positioning of the nitrogen atom in the bi-aryl (UOM-005550 vs UOM-005636) and the increase in size of the aliphatic heterocycle (UOM-005606 vs UOM-005607) resulted in non-selective compounds. Furthermore, the para substitution of the phenyl acetate moiety of S656 with significantly smaller or larger groups led to Class II compounds that lack cyclin K MGD capacity. These modifications include extension of aliphatic heterocycles (UOM-005614 vs UOM-005606), acylation of *p*-amino-phenyl acetate (UOM-005476 vs S656), and the incorporation of smaller functional groups (UOM-005201 vs S656). All of the above suggests that the size of the substitution on the phenyl ring is crucial to establish and stabilize interactions that allow the surfaces of CDK12 and DDB1 to engage in efficient protein-protein interactions to obtain cyclin K MGDs. The substitution of the dimethylaminophenyl acetate moiety of S656 with halogens showcase the above. The larger iodide analog of S656 (UOM-005200) is a Class I compound, while the smaller fluoride and chloride analogs (UOM-005197 and UOM-005198, respectively) are members of Class II. Interestingly, the mid-size bromide analog (UOM-005199) clusters as a mixed-class compound.

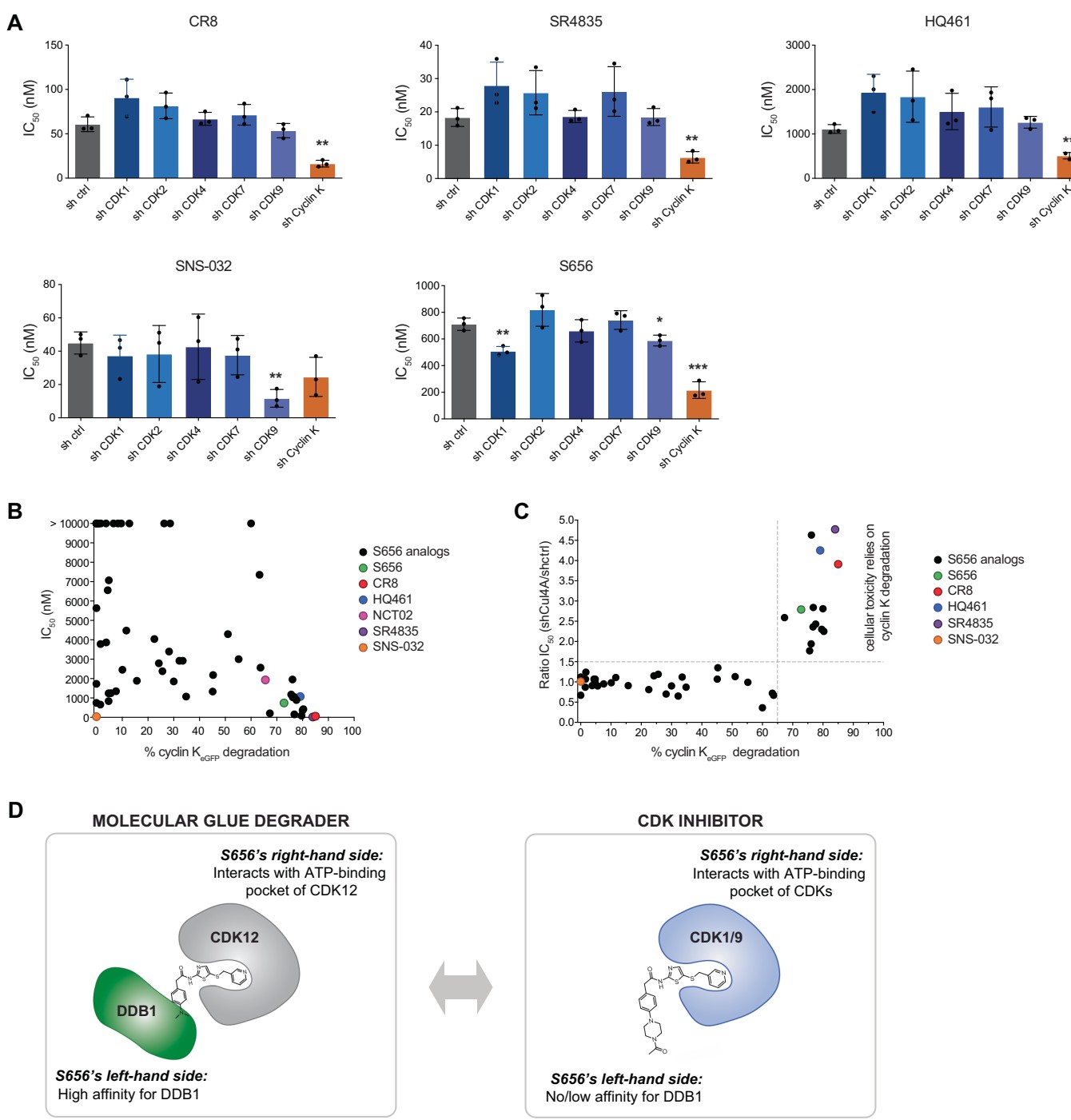

**A** CR8, SR4835, HQ461, SNS-032, S656 bar charts of IC$_{50}$ (nM) across sh ctrl, sh CDK1, sh CDK2, sh CDK4, sh CDK7, sh CDK9, sh Cyclin K

**B** IC$_{50}$ (nM) versus % cyclin K$_{eGFP}$ degradation scatter plot

**C** Ratio IC$_{50}$ (shCul4A/shctrl) versus % cyclin K$_{eGFP}$ degradation scatter plot; cellular toxicity relies on cyclin K degradation

**D**

MOLECULAR GLUE DEGRADER

*S656's right-hand side:* Interacts with ATP-binding pocket of CDK12

CDK12

DDB1

*S656's left-hand side:* High affinity for DDB1

CDK INHIBITOR

*S656's right-hand side:* Interacts with ATP-binding pocket of CDKs

CDK1/9

*S656's left-hand side:* No/low affinity for DDB1

Subsequently, we investigated the role of the aminothiazole core achieving cyclin K MGD activity. Overall, we observe that upon alteration of the hinge interacting triad on the core of the compound, the compounds were rendered inactive. Hence, all compounds showing a substitution on the CH of the aminothiazole, which is part of the aforementioned triad, neither bind to CDKs, nor degrade Cyclin K (UOM-005604, UOM-005603, UOM-005600, UOM-005599, UOM-005598). We also encountered disruption of this binding motif by replacing the CH on the aminothiazole with a

nitrogen (UOM-005601, UOM-005596, UOM-005628; modifications highlighted in grey in Fig. 7A). The CH on the core facilitates stabilization of the molecule towards the hinge and removing it led to a loss of binding of the compounds to CDK. Finally, the replacement of the phenyl acetate moiety of S656 with a adamantyl acetate (UOM-005433), which is likely too bulky and prevents binding to CDKs, leads to an inactive molecule.

We next validated our clustering-based approach of S656 analogs in classes by assessing their potency in a panel of 40

**Figure 6. Hit compound S656 displays both cyclin K degrader function and CDK inhibitory activity.**

(A) Dose response experiment to determine $IC_{50}$ values of the indicated compounds in OCI-AML5 cells expressing shRNAs targeting *CDK1, 2, 4, 7, 9, cyclin K* or a control region (4 days incubation, mean $+/-$ SD, $n = 3$, biological replicates, *t* test, *P* value $= 0.0011$ (sh ctrl vs sh Cyclin K) in CR8, *P* value $= 0.0029$ (sh ctrl vs sh Cyclin K) in SR4835, *P* value $= 0.0011$ (sh ctrl vs sh Cyclin K) in HQ461, *P* value $= 0.0025$ (sh ctrl vs sh CDK9) in SNS-032, *P* value $= 0.0038$ (sh ctrl vs sh CDK1) in S656, *P* value $= 0.0253$ (sh ctrl vs sh CDK9) in S656 and *P* value $= 0.0004$ (sh ctrl vs sh Cyclin K) in S656). (B) Dot plot representation of the $IC_{50}$ values of all S656 analogs (black) and control molecules (colors) in OCI-AML5 cells along with their efficiency to degrade cyclin $K_{eGFP}$ (percentage of degradation after 3 h incubation with 10 µM of compounds, compared to DMSO). Highest tested dose is 10000 nM. (C) Dot plot representation of the ratio of $IC_{50}$ values in OCI-AML5 expressing shRNA control or targeting *CUL4A*, and the percentage of cyclin $K_{eGFP}$ degradation (3 h incubation with 10 µM of compounds, compared to DMSO) for all active ($IC_{50} < 10000$ nM) S656 analogs (black) and control molecules (colors). (D) Working model in which molecules with high affinity for DDB1 behave as cyclin K degraders while modifications of the left-hand side of the molecule, lowering or abolishing the interaction with DDB1, induce cytotoxicity through CDK inhibition. See also Appendix Fig. S5. Source data are available online for this figure.

primary specimens (Appendix Fig. S7A,B). As anticipated, Class I molecules highly correlated with known cyclin K MGDs, while Class II molecules showed a weak correlation, signifying a switch in target engagement. Mixed Class I–II molecules exhibited varying degrees of correlation, likely reflecting their predominant biological activity i.e., whether they more efficiently degrade cyclin K or inhibit CDKs in vivo.

## Best analog UOM-005636 forms an excellent fit in the binding site formed by CDK12 and DDB1

To study the binding mode of selected compounds from different classes, we performed molecular docking studies of UOM-005636 (Class I) and UOM-005197 (Class II) using Glide. Similar to S656 (Fig. 5A), UOM-005636 and UOM-005197 retained the H-bond network with Met816 and the aromatic H-bond to the Glu814 backbone carbonyl showing engagement with the ATP-binding site of CDK12. However, the analysis of surface-area engaged in binding to DDB1 suggests that UOM-005197 (Class II) does not occupy the deep DDB1 pocket formed within the protein complex (Fig. 7B), compared to other analogs that show high degradation activity. Conversely, UOM-005636 (Class I) reveals a perfect fit in the binding site formed by CDK12 and DDB1 (Fig. 7C,D). When compared to our hit S656, which is also reaching crucial interactions on both proteins, UOM-005636 protrudes further into the DDB1 pocket, displaying perfect van der Waals fit and fully occupying the DDB1 binding pocket. The hinge binding motif of UOM-005636 is placed ideally inside the CDK12 binding pocket to establish crucial interactions. Figure 7E shows the overlay of the docked poses of UOM-005636 and S656, suggesting how UOM-005636 is optimized in terms of fit and interaction network. In agreement with these molecular docking results, UOM-005636 shows high degradation of cyclin K and is the most potent and selective analog generated in our SAR study. It is 8 times more potent than S656 in OCI-AML5 cell line ($IC_{50} = 85$ nM versus 743 nM) and in a panel of 40 primary AMLs (median $IC_{50} = 154$ nM versus 1208 nM) (Fig. 7A; Appendix Fig. S7B).

## Discussion

Using a phenotypical screen on genetically diverse primary AML specimens, we identified S656 as a selective anti-tumoral molecule with low toxicity on cord blood cells. We further demonstrated that S656 acts as a cyclin K MGD, mediating the CRL4-dependent proteolysis of cyclin K and destabilization of its CDK12 partner.

Initially, S656's biological significance appeared similar to that of the covalent CDK12/13 inhibitor THZ531. However, its anti-proliferative activity did not correlate with S656 or any of the cyclin K MGDs in a panel of 40 primary AML samples. Although surprising, this suggests that cyclin K loss is not functionally equivalent to the direct inhibition of CDK12/13. Accordingly, Kozicka et al recently reported that transcriptional signatures associated to cyclin K degradation are different from degradation or inhibition of CDK12 (Kozicka et al, 2024). Examples of degradation versus inhibition of receptor tyrosine kinases (RTK) have shown that biological effects can differ. Degraders provide a more sustained pathway inhibition, prevent a "kinome rewiring" effect or avoid reminiscent scaffolding roles of the targeted kinase (Burslem et al, 2018) compared to direct inhibition. This observation alone further warrants the development of selective and potent cyclin K MGD chemical probes (Arrowsmith et al, 2015; Bunnage et al, 2013; Hartung et al, 2023) to study the definite biological impact of cyclin K loss compared to CDK12 inhibition.

The strength of our approach to correlate anti-proliferative activity, especially in primary specimens instead of cell lines, lies in revealing the meaningful functional activity of the compounds. When highly correlated, compounds are intended to share the same molecular target or pathway (Weinstein et al, 1997; Baccelli et al, 2017). As a proof of concept, all cyclin K MGDs tested, despite their structural diversity, were highly correlated. This also suggests that even though CR8 has promiscuous CDK inhibitory activities, its relevant biological activity relies on cyclin K loss rather than CDK inhibition in AML.

Due to the lack of selectivity of S656 towards cyclin K degradation, we performed a SAR study with the aim of understanding the structural features leading to degradation and optimizing S656 cyclin K MGD function. Guided by in silico studies using the available crystal structures of the CDK12-DDB1 complex, we identified that the aminothiazole core of S656 interacts with the hinge region of CDK12 similarly to CR8. Docked poses of S656 in the CDK12-DDB1 complex, showcased the importance of the *para*-dimethylaminophenyl acetate moiety, as it interacts with DDB1 by forming a cation-π interaction between the aromatic ring and Arg928 of DDB1. Hence, the SAR study was focused on modifying the groups that interact with DDB1, which showed that the nature of the functional groups that allows DDB1 recruitment involves a complex balance between steric and electronic effects. Through this exercise we were able to generate two novel selective cyclin K MGDs: UOM-005636 and UOM-005608, showing that both hydrophobic aromatic and

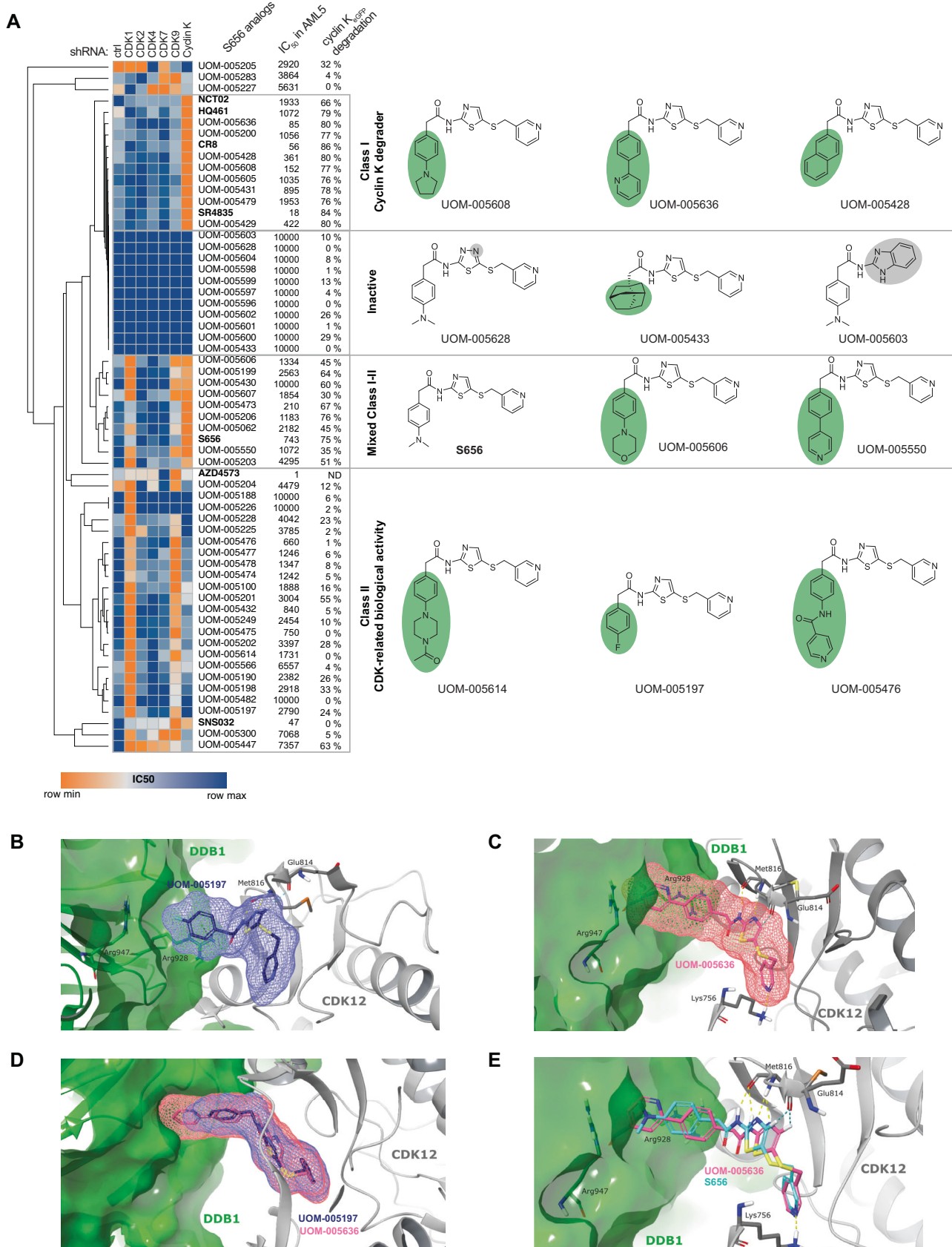

**Figure 7. Minimal modifications in the DDB1 interacting surface optimizes cyclin K degrader function.**

(A) Heatmap clustering S656 analogs and control molecules according to $IC_{50}$ values obtained in OCI-AML5 stably expressing shRNAs targeting *CDK1, 2, 4, 7, 9* as well as *cyclin K*. $IC_{50}$ values in OCI-AML5 (4 days incubation) and the percentage of cyclin $K_{eGFP}$ degradation (3 h, 10 μM, compared to DMSO) are displayed for each analog. For each class of molecules, 3 representative structures are depicted. Relative color scheme uses the minimum and maximum $IC_{50}$ values in each row. Highest dose tested is 10,000 nM. (B–E) 3D representation of the indicated molecule (stick representation) bound to CDK12 (grey ribbons) and DDB1 (green surface representation). The van der Waals surface of the molecules are shown in Mesh to showcase the volume of the molecule. In (B), UOM-005197 forms a cation-π interaction between the phenyl ring and Arg928 of DDB1, but the DDB1 pocket formed by the residues Asn907, Ile909 and Arg947 cannot be filled. Hinge region interactions on CDK12 are conserved. In (C), UOM-005636 shows interaction with the hinge region of CDK12: Met816 provides two backbone interactions, a hydrogen bond donor, and a hydrogen acceptor, respectively, while Glu814 participates in an aromatic H-bond. UOM-005636's Pyridine ring also engages in a H-bond with Lys756. Regarding DDB1, UOM-005636's phenyl ring establishes a cation-π interaction with Arg928 (green dashed line), and its pyridine ring filled DDB1's pocket. (D) Overlay of the 3D representation of UOM-005197 (dark purple) and UOM-005636 (pink) bound to CDK12 and DDB1. UOM-005636 reaches into the pocket formed by DDB1 while UOM-005197 does not enter the DDB1 pocket. (E) Overlay of the 3D representation of S656 (turquoise) and UOM-005636 (pink) bound to CDK12 and DDB1 showing that UOM-005636 reaches further into the DDB1 pocket compared to S656. Hydrogen bonds: yellow dashed line, aromatic H-bonds: turquoise dashed line. See also Appendix Figs. S6 and S7.

aliphatic *para* substitution of the phenyl acetate group of S656 are compatible with selective and potent cyclin K MGDs (Class I compounds). Docking studies suggest that UOM-005636, our most potent and selective analog, is ideal in both, maintaining interactions in the hinge region of CDK12 and filling the DDB1 pocket. Notably, when the groups that interact with the pocket wall of DDB1 contain heteroatoms (e.g. UOM-005550 and UOM-005606), the selectivity as cyclin K MGD is lost, and these compounds display a mixed MGD/CDK inhibition profile. Regarding the DDB1 binding element, if the substituents are either too bulky (e.g. UOM-005614 and UOM-005476) or too small (e.g. UOM-005197), the interactions with DDB1 are impaired, disrupting the formation of the CDK12-DDB1 complex and consequently losing the MGD ability. Our hypothesis is supported by the fact that attaching a PEG linker to the dimethylamine of S656 allowed us to specifically pull down CDK1, 2, and 9, indicating that the molecule, which lost the ability to recruit DDB1, may rather act as a CDK inhibitor.

The structural diversity observed across cyclin K degraders suggests that a broader chemical space can be explored to discover novel MGDs. Overall, we found that for a compound to be a potent and selective cyclin K MGD it must satisfy the following requirements: (a) The CDK12 hinge binding pharmacophore must engage two backbone interactions with CDK12's Met816 through a H-bond donor—H-bond acceptor partner as well as Glu814's backbone carbonyl should be addressed; (b) The DDB1 binding element must interact with Arg928 via a cation-π interaction; (c) The pocket of DDB1 must be fully occupied, preferably with terminal groups that are hydrophobic, while showing aliphatic or aromatic character; and (d) the molecule requires a methylene linker allowing the core and the DDB1 binding element, to adopt an L-shaped conformation. These conclusions fully support those recently described by Kozicka et al using series of cyclin K degraders (Kozicka et al, 2024).

Finally, using an integrated genetic approach to dissect the role and importance of the modulation of various biological targets in specific cellular contexts, especially with promiscuous and non-selective chemotypes, provided invaluable insights on the biological effect of these targets and the structural determinants at play for the functional activity. The understanding of the definitive biological effect of cyclin K loss, using the specific tools developed in our study, warrants promising development and future clinical applications of such degrader molecules.

## Methods

### Reagents and tools table

| Reagent/resource | Reference or source | Identifier or catalog number |
|---|---|---|
| **Experimental models** | | |
| OCI-AML5 (*H. sapiens*) | The University Health Network (Toronto) | N/A |
| OCI-AML1 (*H. sapiens*) | The University Health Network (Toronto) | N/A |
| HEK293T (*H. sapiens*) | ATCC | ATCC-CRL-3216 |
| U2OS (*H. sapiens*) | ATCC | ATCC (HTB-96) |
| **Recombinant DNA** | | |
| Cyclin K - eGFP | Addgene | 169930 |
| pCW-Cas9 | Addgene | 50661 |
| pLX-sgRNA | Addgene | 50662 |
| Extended Knockout (EKO) pooled lentiviral library | Bertomeu et al, 2018 | N/A |
| **Antibodies** | | |
| Cyclin K | Santa Cruz Biotechnology | sc-376371 |
| CDK12 | Cell Signaling Technology | 11973S |
| CDK13 | Santa Cruz Biotechnology | sc-81837 |
| RNA polymerase II phospho-serine2 | Abcam | ab5095 |
| DDB1 | Cell Signaling Technology | 5428S |
| c-MYC | Abcam | ab32072 |
| MCL1 | Cell Signaling Technology | 4572S |
| Alpha-tubulin | Cell Signaling Technology | 2144S |
| GAPDH | Cell Signaling Technology | 97166S |
| Anti-mouse HRP | Jackson ImmunoResearch | 115-035-146 |
| Anti-rabbit HRP | Jackson ImmunoResearch | 111-035-144 |
| Annexin V-Alexa647 | Invitrogen | A23204 |
| Phospho-histone H2A.X (Ser139) | Millipore | 16-193 |
| Cy3-streptavidin | Jackson ImmunoResearch | 016-160-084 |
| **Oligonucleotides and other sequence-based reagents** | | |
| shRNA vectors | This study | Table EV1 |
| PCR primers | This study | Table EV2 |

| Reagent/resource | Reference or source | Identifier or catalog number |
|---|---|---|
| **Chemicals, enzymes and other reagents** | | |
| DMSO | Sigma-Aldrich | D4540 |
| MLN4924 | Adooq Bioscience | A11260 |
| THZ531 | MedChemExpress | HY-103618 |
| S656 | In house synthesis | N/A |
| HQ461 | In house synthesis | N/A |
| CR8 | Sigma-Aldrich | C3249 |
| SR4835 | MedChemExpress | HY-130250 |
| SNS-032 | In house synthesis | N/A |
| NCT-02 | In house synthesis | N/A |
| MG-132 | Adooq Bioscience | A11043 |
| JetPrime transfection reagent | PolyPlus Transfection | 114-15 |
| StemSpan-ACF | StemCell Technology | 9855 |
| RH SCF | Shenandoah Biotechnology | 100-04 |
| RH FLT3L | Shenandoah Biotechnology | 100-21 |
| Hu-TPO | R&D system | 288-TPN |
| Hu LDL | StemCell Technology | 02698 |
| RH IL3 | Shenandoah Biotechnology | 100-80 |
| RH G-CSF | Shenandoah Biotechnology | 100-72 |
| UM729 | In house synthesis | N/A |
| SR1 | StemCell Technology | 72342 |
| CellTiterGlo | Promega | G9241 |
| **Software** | | |
| FlowJo v10 | FlowJo | |
| Prism v6 and v9 | GraphPad Software | |
| R version 4.0 | https://www.r-project.org | |
| **Other** | | |
| CDK1 assay kit | BPS Bioscience | 79597 |
| CDK9/CyclinT kinase assay kit | BPS Bioscience | 79628 |
| CDK12/CyclinK kinase assay kit | BPS Bioscience | 78298 |
| NanoBiT PPI starter systems | Promega | N2014 |
| EasySep Human Cord Blood CD34+ Selection Kit | StemCell Technologies | 18056 |

## Study approval

The Leucegene project is an initiative approved by the Research Ethics Boards of Université de Montréal and Maisonneuve-Rosemont Hospital. All leukemia samples and paired normal DNA specimens were collected and characterized by the Quebec Leukemia Cell Bank after obtaining an institutional Research Ethics

Board–approved protocol with informed consent according to the Declaration of Helsinki. The Quebec Leukemia Cell Bank is a biobank certified by the Canadian Tissue Repository Network. Approval reference numbers: #2018-306 and #2023-4463 from the Université de Montréal and MP-12-2002-366, 01085 from the BCLQ.

## Cytogenetic analyses and cohort definitions

Cytogenetic aberrations and composite karyotypes of the Leucegene cohort were described according to the International System for Human Cytogenomic Nomenclature 2016 guidelines. Complex karyotype was defined as having 3 or more clonal chromosomal abnormalities in the absence of the recurrent genetic abnormalities, including t(8;21), inv(16) or t(16;16), t(9;11), t(6;9), inv(3) or t(3;3) and AML with *BCR-ABL1* (Döhner et al, 2017).

## Cell culture

Cell lines were purchased from the ATCC or donated from collaborators. OCI-AML5 cell line was cultured in αMEM, 10% heat-inactivated FBS supplemented with 10 ng/mL GM-CSF, OCI-AML1, U2OS and HEK293T cell lines in DMEM, 10% heat-inactivated FBS. Cells were maintained at 37 °C in 5% $CO_2$ atmosphere.

## Primary AML sample culture and chemical screens

Freshly thawed primary AML specimens were used for chemical screens. Cryopreserved cells were thawed at 37 °C in IMDM containing 20% FBS and DNase I (100 μg/mL). Cells were resuspended in IMDM supplemented with 15% BIT (BSA, insulin, transferrin; StemCell Technologies), 100 ng/mL SCF, 50 ng/mL FLT3L, 20 ng/mL IL3, 20 ng/mL G-CSF, $10^{-4}$ mol/L b-mercaptoethanol, gentamicin (50 μg/mL), ciprofloxacin (10 μg/mL), SR1 (500 nmol/L) and UM729 (500 nmol/L). Cells were plated in 384-well white plates, 5000 cells per well in 50 μL. Compounds were dissolved in DMSO, diluted in media immediately before use and added to seeded cells at the unique concentration of 1 μM. Control wells received DMSO (0.1%) only. Cell viability was evaluated after 6 days in culture using the CellTiterGlo assay according to the manufacturer's instruction. Percentage of inhibition for dose response curves was calculated as $100 - (100 \times (\text{mean luminescence [compound]/mean luminescence [DMSO]}))$. Dose response curves were generated using nonlinear regression in GraphPad Prism version. For compounds that failed to inhibit AML cell survival/proliferation, $IC_{50}$ values were arbitrarily reported at the highest dose tested (10 μM).

## Human cord blood cell collection and processing

Fresh umbilical cord blood units were collected from consenting donors according to ethically approved procedures at St Justine, Maisonneuve-Rosemont (Montréal, QC, Canada) and Charles Le Moyne (Longueuil, QC, Canada, approval reference number AA-HCLM-16-014) Hospitals. Human CD34+ cells were isolated using the EasySep Human Cord Blood CD34+ Selection Kit through positive magnetic selection. Chemical screen was performed as

described in primary AML with the following modifications: 2000 cord blood cells were seeded, CB media contains StemSpan-ACF supplemented with 100 ng/mL SCF, 100 ng/mL FLT3L, 50 ng/mL TPO, 10 µg/mL LDL, 1× glutamax, gentamicin (50 µg/mL), ciprofloxacin (10 µg/mL), SR1 (500 nmol/L) and UM729 (500 nmol/L).

## Dose-response assays

Dose-response assays in primary AML specimens were performed as in chemical screen (see above) with the exception that compounds were tested in serial dilution (8 dilutions, 1:4, 10 µM down to 0.5 nM). Dose-response assays in OCI-AML5 cell line were conducted similarly with the following modifications: 300 cells per well were seeded in 50 µL of cell line media and cell viability was evaluated after 4 days in culture.

## Compounds

The following compounds were synthetized in house: S656, HQ461 (Lv et al, 2020), SNS-032 (Misra et al, 2004) and NCT02 (Dieter et al, 2021).

## Kinetic solubility measurement

A 40 µL aliquot of a 10 mM DMSO stock solution of S656 was prepared and diluted in αMEM media to achieve a nominal concentration of 200 µM. The samples were incubated with stirring at 1100 rpm for 24 h at room temperature. From the incubation mixture, 20 µL was pipetted, rinsed in acetonitrile (MeCN) for 5 s, then in water for 5 s. Subsequently, 10 µL was discarded, and the remaining 10 µL was transferred to a mixture containing 10 µL of PBS, 980 µL of quenched solution, and 10 µL of DMSO. Analysis was performed using HPLC equipped with a UV-visible detector (HPLC-UV). Chromatography was conducted under reverse-phase conditions with a column temperature of 55 °C on a Waters XSelect HSS T3 column (2.5 µm, 2.1 ×50 mm). The aqueous solvent (solvent A) was ultrapure water with 0.1% formic acid, and the organic eluent (solvent B) was acetonitrile. The injection volume was 4 µL, and the flow rate was set at 0.6 mL/min. The elution program used was as follows: 1/0 min: 95% A, 5% B; 2/2 min: 2% A, 98% B; 3/2.5 min: 2% A, 98% B. The aqueous kinetic solubility of S656 was determined to be 23 µM, calculated from triplicate measurements.

## Whole genome CRISPR/Cas9 deletion screens

As previously described (Moison et al, 2022), we used the Extended Knockout (EKO) pooled lentiviral library developed by Bertomeu et al (Bertomeu et al, 2018) to conduct whole genome CRISPR/Cas9 loss-of-function screen. Briefly, OCI-AML1 EKO cells expressing a doxycycline-inducible Cas9 were cultured in 10% FBS DMEM supplemented with 2 µg/mL doxycycline for a period of 7 days to induce knockouts. The knockout library was maintained in culture 14 more days with exposure to 1 µM of S656 or DMSO (without doxycycline). Genomic DNA was extracted and sgRNA sequenced as described. Synthetic rescue/positive selection and synthetic lethality/negative selection beta scores, as well as statistical significance, were determined using MAGeCK-VISPRMAGeCK-MLE method (Li et al, 2015).

## Immunoblot analysis

Total proteins were extracted using RIPA buffer (20 mM Tris-HCl pH 7.4, 150 mM NaCl, 5 mM MgCl2, 5 mM EGTA, 60 mM β-glycerophosphate, 0.1% NP40, 0.1% Triton X-114, 1 mM DTT) supplemented with protease inhibitors (cOmplete, EDTA-free protease inhibitor tablets), and quantified by the bicinchoninic acid (BCA) method using a BSA standard curve. Proteins were resolved by SDS-PAGE, transferred onto PVDF membrane, blocked with 5% milk and probed with primary (overnight, 4 °C) and secondary (1 h, room temperature) antibodies.

## Cyclin K reporter system

We used the cyclin $K_{eGFP}$ reporter system previously described (Słabicki et al, 2020). Lentiviral vector (Addgene plasmid #169930) was used to infect OCI-AML5 at a MOI of 5 and were selected with puromycin. Cells were seeded in methylcellulose and let grown as colonies for 10 days. Colonies were isolated, expanded and selected based on cyclin $K_{eGFP}$ fluorescence in presence and absence of cyclin K degrader molecules. Selected G7 clone was used for all reported experiments. Fluorescent signal of cyclin $K_{eGFP}$ and mCherry were quantified by flow cytometry (BD LSRII) in each condition and the geometric mean fluorescence intensity (MFI) were calculated using FlowJo (BD Biosciences). The ratio of eGFP to mCherry was normalized to DMSO-treated cells.

## Knockdown experiments

Lentiviral vectors carrying shRNAs targeting *CUL4A* or *CUL4B* gene were generated by cloning shRNA sequences into MNDU vectors comprising miR-E sequences (Fellmann et al, 2013). Control vector (shctrl) contained shRNA targeting Renilla luciferase. Sequences of the 97-mer shRNAs are available in Table EV1. HEK293T cells were transfected with 5 µg lentiviral plasmid, 3.3 µg PAX2 packaging plasmid and 1 µg VSV-G envelope plasmid using 20 µL of JetPrime Transfection reagent, according to manufacturer's directions. Viral supernatants were collected after 48 h, filtered and used to infect cell lines at a multiplicity of infection of 5 in media supplemented with polybrene for 48 h. Infection efficiency, determined by the percentage of GFP$^+$ cells, was monitored by flow cytometry and infected cells were selected with puromycin.

Clonal OCI-AML5 cells expressing an inducible Cas9 (Addgene plasmid #50661 (Wang et al, 2014)) were generated and then infected with lentiviruses for constitutive expression of sgRNAs targeting AAVS1 control region (Addgene plasmid #50662 (Wang et al, 2014)) or DDB1 #1 (GGAAAAGACCAACCTCCTGG); #2 (AAAGGCCATCATAGAGACGC). Cas9 expression is induced by doxycycline (2 µg/mL) and experiments performed subsequently.

## Whole-proteome analysis

Whole proteome was performed as previously described (Moison et al, 2024). We treated 10 million OCI-AML5 cells in triplicate with DMSO or 8 µM S656 for 5 h. Collected cells were washed in PBS and cell pellets lysed in Triton X-100 buffer (10 mM PIPES pH 7.4, 0.5% Triton X-100, 300 mM Sucrose, 100 mM NaCl, 3 mM MgCl2, 0.5 mM EDTA + protease inhibitors). Following MS

analysis, the data were processed using PEAKS X Pro (Bioinformatics Solutions, Waterloo, ON) and a Uniprot human database (20366 entries) with trypsin as the enzyme. Differential Enrichment analysis of Proteomics data (DEP) package (Zhang et al, 2018) in R was used to analyze the data (excluding proteins identified based on a single peptide), including data filtering, normalization, imputation of missing values and statistical testing of differentially expressed proteins.

## Computational methods

### PDB

6TD3 was employed for docking studies (using Glide in Schrödinger, 2022). Chain A (DNA damage-binding protein 1) and B (Cyclin-dependent kinase 12) were prepared using Protein Preparation Workflow in Maestro 13.2.128, Schrödinger. Induced Fit Docking was used to dock the compounds into the active side, keeping the sidechains of residues in the binding site flexible.

### Binding pose metadynamics (BPMD) using Desmond in Schrödinger, 2022

The protein complexes for compounds S656 and UOM-005628 derived from Molecular Docking studies were prepared and solvated for Molecular Dynamics Simulations. Short simulations of 10 ns were repeated 10×, resulting in a reported average as a measure of stability of the ligand during the metadynamics simulations.

### Reported values

PoseScore indicates the average RMSD from the starting pose. Rapid increase in the PoseScore is indicative of ligands that are not in a well-defined energy minimum and therefore the modeled pose might not present a valid binding mode. PersistenceScore (PersScore) is a measure of the H-bond persistence calculated as the fraction of the frames in the last 2 ns of the simulation that have the same hydrogen bonds as the input structure, averaged over all ten repeats.

## Pull-down experiments

Probe compounds were immobilized through their terminal amine on NHS-sepharose beads as previously described (Médard et al, 2015). The prepared beads were then washed in PBS and equilibrated in Triton X-100 buffer. For each replicate, 1–2 mg of total proteins (2 mg/ml) were incubated end-over-end with 25 µl of beads overnight at 4 °C in low protein binding microtubes (Sarstedt). The samples were then washed twice in Triton X-100 buffer, transferred to new microtubes and further washed six times in PBS including another microtube transfer at the last wash. Beads pellets were resuspended in 50 mM ammonium bicarbonate and subjected to reduction, alkylation and tryptic digest as described above. Peptides were separated on an home-made reversed-phase column (150-µm i.d. by 200 mm) with a 56-min gradient from 10 to 30% ACN-0.2% FA and a 600-nl/min flow rate on a Easy nLC-1200 connected to a Exploris 480 (ThermoFisher Scientific, San Jose, CA). Each full MS spectrum acquired at a resolution of 120,000 was followed by tandem-MS (MS-MS) spectra acquisition on the most abundant multiply charged precursor ions for 3 s. Tandem-MS experiments were performed using higher energy

collision dissociation (HCD) at a collision energy of 34%. The data were processed using PEAKS X Pro (Bioinformatics Solutions, Waterloo, ON) and a Uniprot human database (20366 entries). Mass tolerances on precursor and fragment ions were 10 ppm and 0.01 Da, respectively. Fixed modification was carbamidomethyl (C). Variable selected posttranslational modifications were acetylation (N-ter), oxidation (M), deamidation (NQ), phosphorylation (STY). The data were visualized with Scaffold 5.0 (protein threshold, 99%, with at least 2 peptides identified and a false-discovery rate [FDR] of 1% for peptides).

Differential Enrichment analysis of Proteomics data (DEP) package (Zhang et al, 2018) in R was used to analyze the data (excluding proteins identified based on a single peptide), including data filtering, normalization, imputation of missing values and statistical testing of differentially expressed proteins.

## Flow cytometry analysis

For cell survival analysis, 250,000 cells were co-stained with Annexin V-Alexa647 and Propidium Iodide for 15 min in Annexin binding buffer (10 mM HEPES pH 7.4, 14 mM NaCl, 2.5 mM $CaCl_2$). After wash, cells were recorded on BD LSRII cytometer and analyzed with FlowJo v10.

For cell cycle experiments, 500,000 cells in exponential growth phase were incubated in pre-warmed media containing 10 µg/mL Hoechst 33342 during 45 min in incubator (37 °C in 5% $CO_2$). After wash, cells were resuspended in cold media and recorded on BD FACSCelesta cytometer at low flow rate and analyzed with FlowJo v10 (Tree Star).

## CDK1, CDK9 and CDK12 in vitro activity

Kinase activity was assessed using the kinase assay kits from BPS Bioscience for CDK1, CDK9 and CDK12 with Kinase Glo MAX in duplicates. Compounds were tested at a single concentration of 1 µM for CDK1 and CDK9 activity, and in a dilution ranging from 0.64 nM to 50 µM for CDK12 activity. Chemiluminescence was read using the Neo Synergy BioTek plate reader.

## NanoLuc binary technology

Using the NanoBiT PPI MCS starter system vectors, we generated fusion proteins of CDK12 and DDB1 to monitor their interaction in living cells through a structural complementation reporter system for the NanoLuc. The kinase domain of CDK12 was fused to the Large BiT (LgBiT) while DDB1 was fused to the Small Bit (SmBiT), both in N-terminal. The primers used to clone CDK12 and DDB1 in NanoBit vectors by Gibson assembly are as follows: **N-LgBiT-CDK12KD-F:** GTGGGAGTTCCGGTGGTGGCGGGAG CGGAGGTGGAGGCTCGAGCGGTGGAAGACAAACAGAAAG CGACTGG; **N-LgBiT-CDK12KD-STOP-R:** CATGTCTGCTCGA AGCGGCCGGCCGCCCCGACTCTAGAAGATCTGCTAGCTCA TTGTCGCTGACGTCGCCGTTT; **N-SmBiT-DDB1-F:** GTGGAG GCTCGAGCGGTGGAGCTCAGGGGAATTCACAATTGATGTC GTACAACTACGTGG; **N-SmBiT-DDB1-GA-R:** AATGTATCT-TATCATGTCTGCTCGAAGCGGCCGGCCGCCCCGACTCTA-GACTAATGGATCCGAGTTAGCTCC.

Both CDK12-LgBiT and DDB1-SmBiT generated plasmids were then transfected in HEK293 cells using JetPrime transfection

reagent according to the manufacturer's instructions and plated 24 h later in white 384-well plate (20,000 cells /well). After 4 h, 5× D-blue substrate was added to each well, incubated for 5 min at obscurity, prior to addition of compounds of interest at increasing concentrations. Luminescence was monitored for 1 h using the Neo Synergy BioTek plate reader.

## Immunofluorescence

Cells grown on eight-well Ibidi microscopy chamber were pre-extracted 2 min at 4 °C in pre-extraction buffer (25 mM Hepes pH 7.5, 50 mM NaCl, 1 mM EDTA, 3 mM $MgCl_2$, 300 mM sucrose and 0.5% Triton X-100) before fixation in 4% PFA. Samples were then blocked in PBS, 1% bovine serum albumin and stained for 1 h at room temperature with an anti-phospho-histone H2A.X (Ser139) antibody conjugated to biotin (1/1000ème). After washing and staining with Cy3-streptavidin antibody, cells were finally washed and counterstained with DAPI. Following immunostaining, images of a minimum of 49 cells per condition were captured on a Zeiss LSM 700 confocal microscope driven by ZEN software. All images of a given experiment were processed and captured using the same configuration parameters. Images were analyzed using ImageJ software (NIH) to determine the number of γH2AX foci per nucleus. To generate figures, images were processed using Adobe Photoshop and Adobe Illustrator.

## qPCR

RNA was harvested in Trizol (ThermoFisher) and isolated according to manufacturer's protocol and reverse transcribed using MMLV reverse transcriptase and random primers (ThermoFisher). Quantitative PCR was performed using validated assays designed for the Universal Probe Library (Roche) on the Viia7 (Applied Biosystems). qPCR primer sequences are available in Table EV2. Relative quantity of target is normalized to HPRT and compared to normalized expression of control cells.

### Statistical analyses

Statistical differences were determined by unpaired *t* test as indicated in the figure legends. Graphics and statistical analysis were done using GraphPad Prism version 6 or 9. Results having *P* value < 0.05 were considered significant. Statistical differences of $P < 0.05$, $P < 0.01$, $P < 0.001$ and $P < 0.0001$ are depicted as *, **, *** and ****, respectively, in figures.

## Data availability

The mass spectrometry proteomics data have been deposited to the ProteomeXchange Consortium via the PRIDE partner repository and is accessible at https://www.ebi.ac.uk/pride/archive/projects/PXD061850.

The source data of this paper are collected in the following database record: biostudies:S-SCDT-10_1038-S44319-025-00448-y.

## Peer review information

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

## Acknowledgements

The authors wish to thank Muriel Draoui, Mark Wittman and James Carmichael for project coordination, Jean Duchaine, Simon Mathien, Karine Audette and Dominic Salois at the IRIC high-throughput screening platform, as well as the dedicated work of BCLQ members Giovanni d'Angelo, Claude Rondeau and Sylvie Lavallée. We also thank Charles-le Moyne Hospital for providing human umbilical cord blood units. This work was supported by two funds from the Government of Canada through Genome Canada and by the Ministère de l'Economie et de l'Innovation du Québec through Génome Québec—one with additional funding from Bristol-Myers Squibb (LSARP2017 project, reference 13528) and the other with additional funding from RejuvenRx Inc. (GAPP project, reference 6582). JH and GS hold a research chair from Industrielle-Alliance at Université de Montréal and a Bégin-Plouffe chair in blood stem cell chemogenomics of the Faculty of Medicine of Université de Montréal, respectively. BCLQ is supported by grants from the Cancer Research Network of the Fonds de recherche du Québec–Santé. RNA-Seq read mapping and transcript quantification were performed on the supercomputer Briaree from Université de Montréal, managed by Calcul Québec and Compute Canada. This research was enabled in part by support provided by Calcul Québec and the Digital Research Alliance of Canada (alliancecan.ca). J-FS was supported by a Canadian Institutes of Health Research fellowship (MFE-158159).

## Author contributions

**Céline Moison**: Conceptualization; Formal analysis; Supervision; Validation; Investigation; Visualization; Methodology; Writing—original draft; Writing—review and editing. **Rodrigo Mendoza-Sanchez**: Conceptualization; Formal analysis; Supervision; Validation; Investigation; Visualization; Methodology; Writing—original draft; Writing—review and editing. **Deanne Gracias**: Conceptualization; Formal analysis; Validation; Investigation; Methodology; Writing—review and editing. **Doris A Schuetz**: Conceptualization; Formal analysis; Validation; Investigation; Visualization; Methodology; Writing—original draft; Writing—review and editing.

**Jean-François Spinella**: Conceptualization; Formal analysis; Validation; Investigation; Visualization; Methodology. **Simon Girard**: Conceptualization; Investigation; Methodology. **Bounkham Thavonekham**: Conceptualization; Investigation; Methodology. **Jalila Chagraoui**: Conceptualization; Investigation; Methodology. **Aurélie Durand**: Conceptualization; Investigation; Methodology. **Simon Fortier**: Conceptualization; Investigation; Methodology. **Tara MacRae**: Conceptualization; Investigation; Methodology. **Eric Bonneil**: Investigation; Methodology. **Yannick Rose**: Investigation; Methodology. **Nadine Mayotte**: Conceptualization; Investigation; Methodology. **Isabel Boivin**: Conceptualization; Investigation; Methodology. **Pierre Thibault**: Supervision. **Josée Hébert**: Conceptualization; Resources; Data curation; Formal analysis; Supervision; Funding acquisition. **Réjean Ruel**: Conceptualization; Supervision; Writing—review and editing. **Anne Marinier**: Conceptualization; Supervision; Funding acquisition; Writing—original draft; Writing—review and editing. **Guy Sauvageau**: Conceptualization; Supervision; Funding acquisition; Writing—original draft; Writing—review and editing.

Source data underlying figure panels in this paper may have individual authorship assigned. Where available, figure panel/source data authorship is listed in the following database record: biostudies:S-SCDT-10_1038-S44319-025-00448-y.

## Disclosure and competing interests statement

The authors declare no competing interests.

