## [Peer Review File · EMBO Reports]

DDB1 engagement defines the selectivity of S656 analogs for cyclin K degradation over CDK inhibition

Celine Moison, Rodrigo Mendoza-Sanchez, Deanne Gracias, Doris Schuetz, Jean-François Spinella, Simon Girard, Bounkham Thavonekham, Jalila Chagraoui, Aurélie Durand, Simon Fortier, Tara MacRae, Eric Bonneil, Yannick Rose, Nadine Mayotte, Isabel Boivin, Pierre Thibault, Josée Hébert, Réjean Ruel, Anne Marinier, and Guy Sauvageau

Corresponding author(s): *Guy Sauvageau (guy.sauvageau@umontreal.ca)* , *Anne Marinier (anne.marinier@umontreal.ca)*

Review Timeline:

Submission Date:	9th Oct 24
Editorial Decision:	22nd Nov 24
Additional Correspondence from Editor:	26th Nov 24
Revision Received:	29th Jan 25
Editorial Decision:	25th Feb 25
Revision Received:	14th Mar 25
Accepted:	19th Mar 25

Editor: *Martina Rembold*

Transaction Report:

Dear Dr. Sauvageau

Thank you for the submission of your research manuscript to our journal. Three referees agreed to review your manuscript. So far, we have received two referee reports that are copied below. Given that both referees are in fair agreement that you should be given a chance to revise the manuscript, I would like to ask you to begin revising your study along the lines suggested by the referees.

Please note that this is a preliminary decision made in the interest of time, and that it is subject to change should the third referee offer very strong and convincing reasons for this. As soon as we receive the final report on your manuscript, we will forward it to you as well.

Please address all referee concerns in a complete point-by-point response. Acceptance of the manuscript will depend on a positive outcome of a second round of review. It is EMBO Reports policy to allow a single round of revision only and acceptance or rejection of the manuscript will therefore depend on the completeness of your responses included in the next, final version of the manuscript.

We realize that it is difficult to revise to a specific deadline. In the interest of protecting the conceptual advance provided by the work, we recommend a revision within 3 months (February 22nd). Please discuss the revision progress ahead of this time with the editor if you require more time to complete the revisions.

I am also happy to discuss the revision further via e-mail or a video call, if you wish.

*****IMPORTANT NOTE:

We perform an initial quality control of all revised manuscripts before re-review. Your manuscript will FAIL this control and the handling will be delayed IN CASE the following APPLIES:

- 1) A data availability section providing access to data deposited in public databases is missing. If you have not deposited any data, please add a sentence to the data availability section that explains that.
- 2) Your manuscript contains statistics and error bars based on $n=2$. Please use scatter blots in these cases. No statistics should be calculated if $n=2$.

When submitting your revised manuscript, please carefully review the instructions that follow below. Failure to include requested items will delay the evaluation of your revision.*****

- 1) a .docx formatted version of the manuscript text (including legends for main figures, EV figures and tables). Please make sure that the changes are highlighted to be clearly visible.
- 2) individual production quality figure files as .eps, .tif, .jpg (one file per figure). Please download our Figure Preparation Guidelines (figure preparation pdf) from our Author Guidelines pages <https://www.embopress.org/page/journal/14693178/authorguide> for more info on how to prepare your figures.
- 3) a .docx formatted letter INCLUDING the reviewers' reports and your detailed point-by-point responses to their comments. As part of the EMBO Press transparent editorial process, the point-by-point response is part of the Review Process File (RPF), which will be published alongside your paper.
- 4) a complete author checklist, which you can download from our author guidelines (<<https://www.embopress.org/page/journal/14693178/authorguide>>). Please insert information in the checklist that is also reflected in the manuscript. The completed author checklist will also be part of the RPF.
- 5) Please note that all corresponding authors are required to supply an ORCID ID for their name upon submission of a revised

manuscript (<<https://orcid.org/>>). Please find instructions on how to link your ORCID ID to your account in our manuscript tracking system in our Author guidelines (<<https://www.embopress.org/page/journal/14693178/authorguide#authorshipguidelines>>)

6) We replaced Supplementary Information with Expanded View (EV) Figures and Tables that are collapsible/expandable online. A maximum of 5 EV Figures can be typeset. EV Figures should be cited as 'Figure EV1, Figure EV2' etc... in the text and their respective legends should be included in the main text after the legends of regular figures.

7) Before submitting your revision, primary datasets (and computer code, where appropriate) produced in this study need to be deposited in an appropriate public database (see <<https://www.embopress.org/page/journal/14693178/authorguide#dataavailability>>).

The accession numbers and database should be listed in a formal "Data Availability " section (placed after Materials & Method) that follows the model below (see also <<https://www.embopress.org/page/journal/14693178/authorguide#dataavailability>>). Please note that the Data Availability Section is restricted to new primary data that are part of this study.

Data availability

Additional information on source data and instruction on how to label the files are available <<https://www.embopress.org/page/journal/14693178/authorguide#sourcedata>>.

10) Figure legends and data quantification:

- the name of the statistical test used to generate error bars and P values,
 - the number (n) of independent experiments (please specify technical or biological replicates) underlying each data point,
 - the nature of the bars and error bars (s.d., s.e.m.)
- If the data are obtained from n {less than or equal to} 5, show the individual data points in addition to the SD or SEM.
- If the data are obtained from n {less than or equal to} 2, use scatter blots showing the individual data points.

11) Our journal encourages inclusion of *data citations in the reference list* to directly cite datasets that were re-used and obtained from public databases. Data citations in the article text are distinct from normal bibliographical citations and should directly link to the database records from which the data can be accessed. In the main text, data citations are formatted as follows: "Data ref: Smith et al, 2001" or "Data ref: NCBI Sequence Read Archive PRJNA342805, 2017". In the Reference list, data citations must be labeled with "[DATASET]". A data reference must provide the database name, accession number/identifiers and a resolvable link to the landing page from which the data can be accessed at the end of the reference. Further instructions are available at <<https://www.embopress.org/page/journal/14693178/authorguide#referencesformat>>.

12) All Materials and Methods need to be described in the main text using our 'Structured Methods' format. According to this format, the Methods section includes a Reagents and Tools Table (listing key reagents, experimental models, software and relevant equipment and including their sources and relevant identifiers) followed by a Methods and Protocols section describing the methods, ideally using a step-by-step protocol format. The aim is to facilitate adoption of the methodologies across labs. Please download and fill our Reagents and Tools Table template (.docx), which you can find in our author guidelines: <https://www.embopress.org/page/journal/14693178/authorguide#structuredmethods>.

13) As part of the EMBO publication's Transparent Editorial Process, EMBO Reports publishes online a Review Process File to accompany accepted manuscripts. This File will be published in conjunction with your paper and will include the referee reports, your point-by-point response and all pertinent correspondence relating to the manuscript.

Yours sincerely,

=====

Referee #1:

Drs. Sauvageau, Marinier, and their teams have reported the discovery and characterization of the cyclin K molecular glue degrader, S656, in the context of AML treatment. While the mechanisms by which certain CDK12 inhibitors can act as cyclin K degraders are well understood in the TPD space, I find this work to be both interesting and commendable, particularly for its disease-focused drug screening and the chemical biology approaches used to demonstrate cellular effects and dissect the pharmacology of the interplay between degradation and inhibition. The manuscript is quite well-written, with appropriate citations of relevant references. Below, I include several suggestions for the authors to consider to enhance the quality of this manuscript further. The manuscript is suitable for publication after these suggested changes are made.

1. Figures and legends should provide more detailed information on the study conditions. In Figure 1F, over how many days of treatment were the IC50 values obtained? In Figure 2C, which cell line was used, and over how many days of treatment were the IC50 values obtained? In Figure 2F, which cell line was the Western blot performed on? In Figure 2I, over how many days of treatment were the IC50 values obtained? In Figure 2A, which cell line was the Western blot performed on? In Figure 4A, over how many days of treatment were the IC50 values obtained? Figure 5C, over how many days of treatment were the IC50 values obtained?
2. The information in Figure 3A overlaps with that in Figure 2G, so Figure 2G should be removed.

3. In Figure 4A, the print colors used for the compounds are difficult to distinguish. Change the color.
4. The kinetic solubility of S656 should be provided to confirm that its aqueous solubility exceeds the highest dose used in the assays.
5. Before performing the WB analysis, the percentage of cell death at the highest dose over the fixed treatment time should be reported to ensure that protein downregulation is not a result of, but occurs prior to, changes in cell state.
6. In the main text for Figure 2D, the discussion is missing for other significantly downregulated targets such as GNL1, RIOK2, and PMYT1. Are these changes due to direct off-target effects or secondary bystander effects? Is there any evidence that the observed pharmacology of degradation is not influenced by these changes in protein levels?
7. The authors claim that S656 shows sensitivity in certain primary specimens, suggesting its potential for AML treatment. However, there appears to be a disconnect, as degradation characterization is only discussed for OCI-AML5 cells. The authors should provide proteome-wide degradation characterization for at least 2 or 3 responsive specimens.
8. Figure 7 and its relevant sections (Minimal modifications in the DDB1 interacting surface optimize cyclin K degrader function; Best analog UOM-005636 forms an excellent fit in the binding site formed by CDK12 and DDB1) do not add value to the manuscript, as the results and analysis are not meaningful. The authors should remove Figure 7 and the corresponding sections from the main text. Additionally, the differences in degradation activity among the F/C/I analogs are clearly due to the size and polarizability of the halogen atoms. The significance of this aspect is overstated.

Referee #2:

This manuscript by Céline et. al. gives a well-structured analysis of a promising hit compound, S656 against the aggressive AML. Using high-throughput screening of a 10,000-compound library at a concentration of 1 μ M, the authors identified 12 potent candidates and profiled their cytotoxic effects in cell lines and primary AML specimens. Of these, using a CRSPR-Cas9 approach with an extended-knockout (EKO) library of sgRNAs, they identified the hit, S656 demonstrating its mechanism of action via dependency on the Cullin4-RING E3 ubiquitin ligase (CRL4) complex, CUL4A/B-RBX1-DDB1. S656 appears to induce the degradation of cyclin K, a key transcriptional regulator. CDK12-Cyclin K complex, a recently suggested oncotherapy target, is disrupted through inhibition at the ATP-binding site of CDK12, impairing cellular migration, cell cycle progression, and other critical functions. This mechanism, as the authors note, highlights the therapeutic potential of targeting CDK12-Cyclin K in cancer treatment.

Similar to the findings of Ślabicki, M. et al. The CDK inhibitor CR8 acts as a molecular glue degrader that depletes cyclin K. *Nature* 585, 293-297 (2020), the authors demonstrated that S656 also functions as a molecular glue degrader, binding to the CDK12-Cyclin K complex and recruiting the DDB1 component of the CRL4 complex. This interaction facilitates CRL4-mediated polyubiquitination and subsequent proteasomal degradation of Cyclin K. By depleting Cyclin K, S656 suppresses CDK12-Cyclin K-mediated phosphorylation of Serine 2 of RNA polymerase II, disrupting the expression of DNA damage response genes and thereby interfering with tumour progression.

The authors demonstrate that short-term treatment of an AML cell line with S656 led to significant Cyclin K depletion and subsequent destabilization of CDK12. This effect was reversed with the proteasome inhibitor MG132, indicating a proteasome-mediated degradation pathway. Additionally, the suppression of DDB1 and other CRL4 complex components reduced S656 cytotoxicity and prevented Cyclin K degradation. The authors suggest a mechanism in which the compound's activity relies on DDB1 to drive Cyclin K degradation, thereby inhibiting cell survival - an effect distinct from the CDK12 inhibitor THZ531, which did not replicate the observed effect.

The authors further show that Cyclin K depletion by S656 suppresses phosphorylation of Serine 2 on RNA polymerase II and reduces the mRNA transcript levels of key DNA damage response genes. This depletion of Cyclin K also disrupts cell cycle progression from G1 to S-phase and promotes apoptosis, including a reduction in the expression of anti-apoptotic Mcl1 protein. In silico analyses indicate a strong binding affinity of S656 to the kinase domain of CDK12, alongside an interaction with DDB1, which appears to disrupt the CDK12-Cyclin K complex while promoting a CDK12-DDB1 interaction. Among the analogues of S656 tested, a variety of inhibitory mechanisms were observed across AML lines, with reduced potency in Cyclin K degradation suggesting possible non-CRL4-dependent mechanisms. The authors hypothesize that potent analogues lacking Cyclin K degradation activity may act primarily as CDK inhibitors instead. They identify UOM-005636 as a particularly potent analogue, demonstrating increased cytotoxicity against AML specimens, although data on selectivity against normal cells is not included in this report.

Overall, this manuscript presents a comprehensive, collaborative investigation, that addresses the authors' hypotheses from multiple perspectives. While it offers valuable insights and well-detailed findings, a few recommendations provided below may enhance clarity and strengthen the authors' conclusions.

1. The assertion of S656's selectively cytotoxicity against cancer cells is not sufficiently supported by the data presented. With less than 20% cytotoxicity demonstrated across 56 AML specimens and the absence of cytotoxic effects shown in only two normal CD34+ cord blood cells, the findings do not provide enough statistical power neither does it represent a comprehensive comparison. Using only two normal cells as the baseline for toxicity contrasts with 56 AML specimens lacks robustness and fails to establish a reliable measure of selective cytotoxicity. To justify selective cytotoxicity claims, a broader range of normal cells would need to be tested alongside the cancer cells.
2. The authors demonstrate S656 effect as early as 3 hours, depleting Cyclin K (Figure 2 D-E), it would be reasonable to anticipate some observable cell cycle disruptions within the initial early - 12 hours, particularly regarding G1/S phase progression. The authors report, however, that significant cell cycle changes only become apparent after 24 hours. Could there

be a compensatory delay or rescue effect via Cyclin D-CDK4/6, another complex involved in G1/S phase regulation through alternate pathways [Topacio, B. et. al. Cyclin D-Cdk4,6 Drives Cell-Cycle Progression via the Retinoblastoma Protein's C-Terminal Helix. *Mol Cell* 74(4):758-770 (2019)]? This may explain the delayed onset of the cell cycle effects despite an early Cyclin K depletion.

3. It may be worthwhile to investigate whether the CDK12/13 inhibitor THZ531 could synergize with S656 or analogues like UOM-005636 that specifically degrade Cyclin K, in combination cytotoxicity assays, assessing selective suppression of AML viability over normal cells. Both THZ531 and S656 / UOM-005636 disrupt the CDK12-Cyclin K complex - THZ531 by inhibiting CDK12, and S656 / UOM-005636 by degrading Cyclin K. Similar synergistic effects were observed by Katharina et al., where THZ531 promoted prostate cancer cell sensitivity to androgen deprivation therapy [Frei, K et. al. Inhibition of the Cyclin K-CDK12 complex induces DNA damage and increases the effect of androgen deprivation therapy in prostate cancer. *Int J Cancer* 154(6):1082-1096 (2024)].

Dear Dr. Sauvageau

We have now received the third report for your manuscript (Referee #3), which I herewith forward to you, as promised in my earlier mail. Since also this referee is very positive about your study and lists only minor concerns, I reiterate my invitation to revise your study for our journal. Please address the concerns from all three referees in your revision.

Kind regards,

Martina Rembold, PhD
Senior Editor
EMBO Reports

=====

Referee #3

Moison et al., report a new molecular glue degrader (MDG) that targets the CDK12-Cyclin K complex for degradation. The compound (S656) was identified using a high throughput screen in primary cells from AML patients and this showed good selectivity over normal blood cells (CD34+). The authors have extensively characterised the molecule's mode of action and shown through reporter and biophysical assays that the S656 series is consistent with a molecular glue degrader function, and this activity is dependent on the DDB1-CRL4 complexes. They go on to provide molecular models based on docking and MD simulations that are again consistent with the compounds engaging both CDK12 and DDB1, similar to how other CDK12/Cyclin K degraders function. They have effectively used this modelling approach to improve the potency and functional activity of S656 derivatives in cells.

Overall the study is rigorous and the data are of high quality. The results are exciting and interesting from the small molecule discovery and development of MGDs angle, as well as the potential for future therapeutics.

It is a strong candidate for publication in EMBO Reports and will appeal to a wide group of people and audiences (e.g. those interested in drug discovery, event-driven pharmacology, molecular modelling and cancer biology).

I have only minor comments that the authors should address if invited for a resubmission.

Fig. 2I -- What are the primary data for IC50 calculations? It would be useful if the authors showed a representative experiment for at least DMSO control and S656? This could be added in the Supplementary material since this is an assay used extensively in the paper (e.g. Fig. 6, S2 etc.,)

Molecular modelling panels -- The authors could improve these by increasing the surface transparency to better see side chain residues (e.g. it is hard to see Arg side chains in Fig. 5A). Moreover, it is not clear why hydrogens are shown for Lys756 but not others. Either show for all side chains or remove from Lys756. I recommend removing hydrogens from side chain residues unless there is a particular reason for showing them. Please address these minor details for all structure modelling figures to improve readability (also increase font size for some labels).

The authors don't define the error bars in Figure S3A. Similarly, they don't mention if the results shown in Fig. S5 are from a single experiment or if error bars are missing. Please check all figures and be consistent throughout.

Dear Editor,

Thank you for your letter. We want to thank the reviewers for the effort and time invested in our manuscript. Please find below a point-by-point response to their comments and recommendations along with a revised manuscript (changes made are highlighted in red) that hopefully addresses all the key points raised during the review process.

*Sincerely,
Guy Sauvageau*

Referee #1:

Drs. Sauvageau, Marinier, and their teams have reported the discovery and characterization of the cyclin K molecular glue degrader, S656, in the context of AML treatment. While the mechanisms by which certain CDK12 inhibitors can act as cyclin K degraders are well understood in the TPD space, I find this work to be both interesting and commendable, particularly for its disease-focused drug screening and the chemical biology approaches used to demonstrate cellular effects and dissect the pharmacology of the interplay between degradation and inhibition. The manuscript is quite well-written, with appropriate citations of relevant references. Below, I include several suggestions for the authors to consider to enhance the quality of this manuscript further. The manuscript is suitable for publication after these suggested changes are made.

1. Figures and legends should provide more detailed information on the study conditions. In Figure 1F, over how many days of treatment were the IC₅₀ values obtained? In Figure 2C, which cell line was used, and over how many days of treatment were the IC₅₀ values obtained? In Figure 2F, which cell line was the Western blot performed on? In Figure 2I, over how many days of treatment were the IC₅₀ values obtained? In Figure 2A, which cell line was the Western blot performed on? In Figure 4A, over how many days of treatment were the IC₅₀ values obtained? Figure 5C, over how many days of treatment were the IC₅₀ values obtained?

Additional details about the experimental conditions have been added in the figure legends to improve clarity and provide better context.

2. The information in Figure 3A overlaps with that in Figure 2G, so Figure 2G should be removed.

Figure 2G demonstrates that the loss of cyclin K is prevented by MLN4924 and MG132 treatment, while Figure 3A highlights the depletion of RNA Pol II p-Ser2. Although cyclin K is also shown in Figure 3A, this serves as a control for the experiment. While this might appear redundant, we believe it is essential to retain both figures to underscore the strong reproducibility of the experiments, which we believe is important.

3. In Figure 4A, the print colors used for the compounds are difficult to distinguish. Change the color.

The colors have been adjusted to improve clarity and make the data presentation more clear.

4. The kinetic solubility of S656 should be provided to confirm that its aqueous solubility exceeds the highest dose used in the assays.

We determined the kinetic solubility of S656 to be 23 μ M, which exceeds the highest dose tested in all our assays. This data has been incorporated into Materials and Methods for reference.

5. Before performing the WB analysis, the percentage of cell death at the highest dose over the fixed treatment time should be reported to ensure that protein downregulation is not a result of, but occurs prior to, changes in cell state.

This is a very good point. Please note that WB analyses were performed after short-time exposure (5 hours) to cyclin K degrader molecules. At this time point, even at high concentration, the degrader molecules showed limited induction of cell death, as evidenced by the absence of Annexin V / PI positive cells (see below, late apoptosis). In addition, under these conditions, no more than 17% of cells were observed in early apoptosis (Annexin V positive).

6. In the main text for Figure 2D, the discussion is missing for other significantly downregulated targets such as GNL1, RIOK2, and PMYT1. Are these changes due to direct off-target effects or secondary bystander effects? Is there any evidence that the observed pharmacology of degradation is not influenced by these changes in protein levels?

Proteome-wide mass spectrometry analysis in Figure 2D was conducted using a high concentration of S656 (8 μ M) for 5 hours to identify proteins selectively degraded by this molecule. While CCNK (cyclin K) appears as the most downregulated protein, others including GNL1 (G protein Nucleolar 1), RIOK2 (RIO Kinase 2) and PMYT1 (Protein Kinase, membrane Associated Tyrosine/Threonine 1) were also downregulated.

Additional proteome-wide mass spectrometry analyses were performed using another cell line model (OCI-AML2), where cells were exposed to 2 or 6 μ M of S656 for 4 hours. These results consistently demonstrated depletion of CCNK, as well as CDK12 and CDK13, while GNL1,

RIOK2 or PMYT1 were not among the most downregulated proteins (see volcano plots below). Notably, higher concentrations of S656 resulted in a greater number of downregulated proteins, likely reflecting secondary effects of cyclin K loss. This is consistent with the role of the Cyclin K/CDK12 complex in regulating transcriptional elongation across many genes. These observations suggest that GNL1, RIOK2 and PMYT1 are not directly involved in S656-mediated cyclin K degradation.

Figure for referees not shown.

7. The authors claim that S656 shows sensitivity in certain primary specimens, suggesting its potential for AML treatment. However, there appears to be a disconnect, as degradation characterization is only discussed for OCI-AML5 cells. The authors should provide proteome-wide degradation characterization for at least 2 or 3 responsive specimens.

As demonstrated above, cyclin K depletion was observed in various cell lines, including OCI-AML2 cells. While the limited availability of primary AML samples precluded a proteome-wide analysis, we confirmed cyclin K degradation through Western blot analysis in three different primary specimens. Thawed cells were treated with 5 μ M of S656 for 5 hours prior to protein analysis. The results revealed reproducible depletion of cyclin K protein, and are now included in the Figure S2A.

8. Figure 7 and its relevant sections (Minimal modifications in the DDB1 interacting surface optimize cyclin K degrader function; Best analog UOM-005636 forms an excellent fit in the binding site formed by CDK12 and DDB1) do not add value to the manuscript, as the results and analysis are not meaningful. The authors should remove Figure 7 and the corresponding sections from the main text. Additionally, the differences in degradation activity among the F/Cl/I analogs are clearly due to the size and polarizability of the halogen atoms. The significance of this aspect is overstated.

Our research operates at the intersection of biology and medicinal chemistry, and this synergy is fundamental to our approach. Figure 7 exemplifies this interplay by providing a comprehensive analysis of the S656 analogs, linking chemical modifications with biological outcomes. This holistic perspective allowed us to identify critical features necessary for DDB1 recruitment and enhanced activity, providing key insights into the structure-activity relationships and the rational design of novel molecular glue degraders targeting cyclin K.

In light of your feedback and to better communicate the value of these results, we have revised the relevant sections in the manuscript to highlight the importance and context of our findings more effectively. These changes aim to align the analysis more closely with the manuscript's core objectives, ensuring that the significance of this work is clear to the reader.

Referee #2:

This manuscript by Céline et. al. gives a well-structured analysis of a promising hit compound, S656 against the aggressive AML. Using high-throughput screening of a 10,000-compound library at a concentration of 1 μ M, the authors identified 12 potent candidates and profiled their cytotoxic effects in cell lines and primary AML specimens. Of these, using a CRSPR-Cas9 approach with an extended-knockout (EKO) library of sgRNAs, they identified the hit, S656 demonstrating its mechanism of action via dependency on the Cullin4-RING E3 ubiquitin ligase (CRL4) complex, CUL4A/B-RBX1-DDB1. S656 appears to induce the degradation of cyclin K, a key transcriptional regulator. CDK12-Cyclin K complex, a recently suggested oncotherapy target, is disrupted through inhibition at the ATP-binding site of CDK12, impairing cellular migration, cell cycle progression, and other critical functions. This mechanism, as the authors note, highlights the therapeutic potential of targeting CDK12-Cyclin K in cancer treatment. Similar to the findings of Ślabicki, M. et al. The CDK inhibitor CR8 acts as a molecular glue degrader that depletes cyclin K. *Nature* 585, 293-297 (2020), the authors demonstrated that S656 also functions as a molecular glue degrader, binding to the CDK12-Cyclin K complex and recruiting the DDB1 component of the CRL4 complex. This interaction facilitates CRL4-mediated polyubiquitination and subsequent proteasomal degradation of Cyclin K. By depleting Cyclin K, S656 suppresses CDK12-Cyclin K-mediated phosphorylation of Serine 2 of RNA polymerase II, disrupting the expression of DNA damage response genes and thereby interfering with tumour progression.

The authors demonstrate that short-term treatment of an AML cell line with S656 led to significant Cyclin K depletion and subsequent destabilization of CDK12. This effect was reversed with the proteasome inhibitor MG132, indicating a proteasome-mediated degradation

pathway. Additionally, the suppression of DDB1 and other CRL4 complex components reduced S656 cytotoxicity and prevented Cyclin K degradation. The authors suggest a mechanism in which the compound's activity relies on DDB1 to drive Cyclin K degradation, thereby inhibiting cell survival - an effect distinct from the CDK12 inhibitor THZ531, which did not replicate the observed effect.

The authors further show that Cyclin K depletion by S656 suppresses phosphorylation of Serine 2 on RNA polymerase II and reduces the mRNA transcript levels of key DNA damage response genes. This depletion of Cyclin K also disrupts cell cycle progression from G1 to S-phase and promotes apoptosis, including a reduction in the expression of anti-apoptotic Mcl1 protein. In silico analyses indicate a strong binding affinity of S656 to the kinase domain of CDK12, alongside an interaction with DDB1, which appears to disrupt the CDK12-Cyclin K complex while promoting a CDK12-DDB1 interaction. Among the analogues of S656 tested, a variety of inhibitory mechanisms were observed across AML lines, with reduced potency in Cyclin K degradation suggesting possible non-CRL4-dependent mechanisms. The authors hypothesize that potent analogues lacking Cyclin K degradation activity may act primarily as CDK inhibitors instead. They identify UOM-005636 as a particularly potent analogue, demonstrating increased cytotoxicity against AML specimens, although data on selectivity against normal cells is not included in this report.

Overall, this manuscript presents a comprehensive, collaborative investigation, that addresses the authors' hypotheses from multiple perspectives. While it offers valuable insights and well-detailed findings, a few recommendations provided below may enhance clarity and strengthen the authors' conclusions.

1. The assertion of S656's selectively cytotoxicity against cancer cells is not sufficiently supported by the data presented. With less than 20% cytotoxicity demonstrated across 56 AML specimens and the absence of cytotoxic effects shown in only two normal CD34+ cord blood cells, the findings do not provide enough statistical power neither does it represent a comprehensive comparison. Using only two normal cells as the baseline for toxicity contrasts with 56 AML specimens lacks robustness and fails to establish a reliable measure of selective cytotoxicity. To justify selective cytotoxicity claims, a broader range of normal cells would need to be tested alongside the cancer cells.

Our study focuses on acute myeloid leukemia (AML), using normal CD34+ cord blood cells as control for our experiments in the discovery screen. The criteria for prioritizing hit compounds among the 10,000 molecules tested were based on selective inhibition across the AML specimens. Specifically, we prioritized compounds that showed minimal inhibition in the two tested cord blood samples and demonstrated low percentage inhibition in certain primary AML specimens while exhibiting strong inhibition in others.

For comparison, we included below the inhibitory profiles of non-selective compounds obtained in this primary screen: Oligomycin (10 nM), 6-mercaptopurine (1 μ M), Paclitaxel (50 nM) and Panobinostat (50 nM), which all displayed broadly cytotoxic effects. We contrast these profiles with that of S656.

Discovery screen

Additionally, we performed dose-response assays on purified CD34+ cells from cord blood, resulting in an IC_{50} of 1309 nM (see dose-response curve on the right). This value is above 67% of the IC_{50} values observed in primary AML specimens, suggesting that a therapeutic window exist for a large proportion of the AML specimens.

Other groups have investigated the cytotoxicity of cyclin K molecular degraders in normal cells. For instance, Dieter et al. (PMID34289372) reported that NCT02 was inactive on normal primary fibroblasts compared to colorectal cancer cells. Additionally, they demonstrated that the cyclin K degrader molecule SR4835, administered orally in mice (20 mg/kg, 5 days per week), effectively inhibited tumor growth in vivo while being well tolerated in animals.

2. The authors demonstrate S656 effect as early as 3 hours, depleting Cyclin K (Figure 2 D-E), it would be reasonable to anticipate some observable cell cycle disruptions within the initial early - 12 hours, particularly regarding G1/S phase progression. The authors report, however, that significant cell cycle changes only become apparent after 24 hours. Could there be a compensatory delay or rescue effect via Cyclin D-CDK4/6, another complex involved in G1/S phase regulation through alternate pathways [Topacio, B. et. al. Cyclin D-Cdk4,6 Drives Cell-Cycle Progression via the Retinoblastoma Protein's C-Terminal Helix. Mol Cell 74(4):758-770 (2019)]? This may explain the delayed onset of the cell cycle effects despite an early Cyclin K depletion.

As mentioned, the Cyclin K GFP reporter assay used in our paper (Figure 2E) documented cyclin K degradation after 3 hours of exposure (Figure 2E). The Nanobit assay further revealed that interaction between DDB1 and CDK12 occurs within minutes after the addition of S656, or other cyclin K molecular glues (Figure 5F). However, we anticipated that the cellular effects of Cyclin K depletion take time, as its impact is primarily mediated through the alteration of transcriptional elongation, including genes involved in DNA damage response. For this reason, we began monitoring the effects of S656 treatment on cell cycle progression and apoptosis after 24 hours (Figure 3C and D). As pointed out, the effects may manifest earlier, and to explore this, we analyzed the cell cycle after 14 hours of exposure to S656 (at concentrations of 3 and 5 μ M), showing indeed cell cycle defects at this early time point (see below).

3. It may be worthwhile to investigate whether the CDK12/13 inhibitor THZ531 could synergize with S656 or analogues like UOM-005636 that specifically degrade Cyclin K, in combination cytotoxicity assays, assessing selective suppression of AML viability over normal cells. Both THZ531 and S656 / UOM-005636 disrupt the CDK12-Cyclin K complex - THZ531 by inhibiting CDK12, and S656 / UOM-005636 by degrading Cyclin K. Similar synergistic effects were observed by Katharina et al., where THZ531 promoted prostate cancer cell sensitivity to androgen deprivation therapy [Frei, K et. al. Inhibition of the Cyclin K-CDK12 complex induces DNA damage and increases the effect of androgen deprivation therapy in prostate cancer. *Int J Cancer* 154(6):1082-1096 (2024)].

Indeed, THZ531 is a covalent CDK12/13 inhibitor that specifically binds to the kinase pocket of these CDKs. As reported in the literature, we also demonstrated that S656 binds to the kinase pocket of CDK12, and that pre-incubation with THZ531 competes with S656-mediated Cyclin K degradation (see Figure 5E).

We performed a synergistic interaction assay between THZ531 and S656 or UOM-005636 in OCI-AML5 and U937 cell lines (see below). Our results show moderate to very low synergistic interactions between THZ531 and our cyclin K degrader molecules. This is likely due to both drugs competing for the same binding site, rather than acting through complementary pathways. In contrast, the combination therapy of THZ531 and androgen deprivation therapy, as described by Frei et al., relies on the inhibition of two distinct pathways, ultimately synergizing to exert anti-proliferative effects.

Referee #3

Moison et al., report a new molecular glue degrader (MDG) that targets the CDK12-Cyclin K complex for degradation. The compound (S656) was identified using a high throughput screen in primary cells from AML patients and this showed good selectivity over normal blood cells (CD34+). The authors have extensively characterised the molecule's mode of action and shown through reporter and biophysical assays that the S656 series is consistent with a molecular glue degrader function, and this activity is dependent on the DDB1-CRL4 complexes. They go on to provide molecular models based on docking and MD simulations that are again consistent with the compounds engaging both CDK12 and DDB1, similar to how other CDK12/Cyclin K degraders function. They have effectively used this modelling approach to improve the potency and functional activity of S656 derivatives in cells.

Overall the study is rigorous and the data are of high quality. The results are exciting and interesting from the small molecule discovery and development of MGDs angle, as well as the potential for future therapeutics.

It is a strong candidate for publication in EMBO Reports and will appeal to a wide group of people and audiences (e.g. those interested in drug discovery, event-driven pharmacology, molecular modelling and cancer biology).

I have only minor comments that the authors should address if invited for a resubmission.

Fig. 2I -- What are the primary data for IC₅₀ calculations? It would be useful if the authors showed a representative experiment for at least DMSO control and S656? This could be added in the Supplementary material since this is an assay used extensively in the paper (e.g. Fig. 6, S2 etc.,).

For all IC₅₀ calculation reported in the paper, we determined the percentage of inhibition using the following formula: $100 - (100 \times (\text{mean luminescence [compound]} / \text{mean luminescence [DMSO]})$, for each dose. Dose response curves were then generated using nonlinear regression in GraphPad Prism version 6.01. IC₅₀ values represent the concentration of the molecule required to achieve 50% inhibition of the luminescence intensity. As a representative example, we included dose-response curves obtained for the S656 molecule in 3 primary samples tested in the validation screen and showing a range of IC₅₀ values (see below and in Figure S1D).

Molecular modelling panels -- The authors could improve these by increasing the surface transparency to better see side chain residues (e.g. it is hard to see Arg side chains in Fig. 5A). Moreover, it is not clear why hydrogens are shown for Lys756 but not others. Either show for all side chains or remove from Lys756. I recommend removing hydrogens from side chain residues unless there is a particular reason for showing them. Please address these minor details for all structure modelling figures to improve readability (also increase font size for some labels).

Thank you for the insightful feedback. We have adjusted the surface transparency to make the residues more visible, removed the mesh surface of compound S656 to ensure better visibility, and made the ribbons on CDK12 transparent. This should allow for better visibility of important side chains and interacting residues. Residue Lys756 is shown with its polar hydrogens, as those are important, acting as hydrogen bond donors. We have also included the polar hydrogens to other residues shown in Figure 5A to maintain consistency in the representation. Font size has been increased to facilitate readability.

The authors don't define the error bars in Figure S3A. Similarly, they don't mention if the results shown in Fig. S5 are from a single experiment or if error bars are missing. Please check all figures and be consistent throughout.

Errors bars and details on the bar graphs, including the display of individual data points, have been corrected throughout the Figures. In Figure S3A, error bars represent the mean \pm SD ($n=3$). For Figure S5, which presents the qPCR validation of shRNAs, the data were obtained in duplicate, and this has now been highlighted in both the Figure and the corresponding legend.

Dear Dr. Sauvageau

Thank you for the submission of your revised manuscript. It was sent to former referee #2 who considered all concerns adequately addressed and supported publication noting in the summary evaluation table "Is suitable for publication in EMBO reports without revision."

Browsing through the manuscript myself, I noticed a few editorial things that we need before we can proceed with the official acceptance of your study.

- Please reduce the number of keywords to 5.
- Please update the 'Conflict of interest' paragraph to our new 'Disclosure and competing interests statement'. For more information see <https://www.embopress.org/page/journal/14693178/authorguide#conflictsofinterest> and please place it after the Acknowledgments.
- Regarding the Author Contributions, we now use CRediT to specify the contributions of each author in the journal submission system. Therefore, please remove the Author Contributions from the manuscript file and make sure that the author contributions in our online manuscript tracking system are correct and up-to-date. The information you specified in the system will be automatically retrieved and typeset into the article. You can enter additional information in the free text box provided, if you wish.
- Please remove the statement "Data and materials availability: All data are available in the manuscript or in the supplementary materials." And provide a "Data availability" paragraph instead, which is placed at the end of the methods section. Please deposit the proteomics data in a public repository and refer to the data in the Data availability paragraph ((suggested wording: "The [structural coordinates | microarray | mass spectrometry] data from this publication have been deposited to the [name of the database] database [URL] and assigned the identifier [accession | permalink | hashtag].").
- Studies involving human participants: you have clearly documented that leukemia and human cord blood cell samples were obtained after approval by an ethics board and with informed consent, which is good. We usually also ask for the reference number of the approval(s), which I kindly ask you to add.
- Author checklist: please complete the information in C6 (manuscript ID#).
- Funding information: please enter all funders in the online submission system. Please do not use the Comments box for this but the list above. The production team retrieves information only from the separate entries in the system. All the funders acknowledged in the manuscript file need to be provided as separate entries in the system (More Funders button).
- Table EV1-EV5 are datasets and need to be updated to Dataset EV1-EV5 in all places. The legends for these datasets need to be removed from the Appendix file and each should be provided in the corresponding Excel file: in the first row for EV tables and as a separate sheet/tab for datasets.
- Table EV6 and EV7 should be called Table EV1 and EV2. The legend should be provided in the corresponding Excel file, in the first row of the table. Note that these two tables could also be incorporated into the Reagents and Tools table.
- Appendix: Please provide a table of content that lists each Appendix Figure with page numbers. Please remove the information regarding the EV tables and add the descriptive legends as a separate tab in the respective .xls files instead (see above).
- Can the 'Synthetic chemistry supporting information' be part of Appendix file? If you prefer to publish it as an individual Word file, the nomenclature would need to be updated since we do not use the nomenclature "Supplementary". I am happy to discuss different options further.
- Appendix Figure S2, S3, S5: please define the nature of 'n' (technical, biological) in the legend.
- Materials and Methods should be Methods
- There was one more comment regarding the figure legends that seems to need your attention: Please indicate what */ **/ ***/ **** represents; if this represents p value(s), please indicate the exact p value in the legend(s) of figure(s) 2C, E, I; 3B, E; 6A.
- Please use present tense when you describe your findings in the abstract.
- Finally, EMBO Reports papers are accompanied online by
A) a short (1-2 sentences) summary of the findings and their significance,

B) 2-3 bullet points highlighting key results and

C) a schematic summary figure that provides a sketch of the major findings (not a data image).

Please provide the summary figure as a separate file in PNG or JPG format at a size of 550x300-600 pixels (width x height).

Please note that the size is rather small and that text needs to be readable at the final size. Please send us this information along with the revised manuscript.

With kind regards,

Martina Rembold, PhD

Senior Editor

EMBO reports

=====

Referee #2

"Is suitable for publication in EMBO reports without revision"

All editorial and formatting issues were resolved by the authors.

Guy Sauvageau
University of Montreal
2950, chemin de Polytechnique
Québec H3T1J4
Canada

Dear Dr. Sauvageau,

I am very pleased to accept your manuscript for publication in the next available issue of EMBO reports. Thank you for your contribution to our journal.

Kind regards,
